# FastCAV: Efficient Computation of Concept Activation Vectors for Explaining Deep Neural Networks

**Laines Schmalwasser** [1 2]  **Niklas Penzel** [2]  **Joachim Denzler** [2]  **Julia Niebling** [1]

## Abstract

Concepts such as objects, patterns, and shapes are how humans understand the world. Building on this intuition, concept-based explainability methods aim to study representations learned by deep neural networks in relation to human-understandable concepts. Here, Concept Activation Vectors (CAVs) are an important tool and can identify whether a model learned a concept or not. However, the computational cost and time requirements of existing CAV computation pose a significant challenge, particularly in large-scale, high-dimensional architectures. To address this limitation, we introduce FastCAV, a novel approach that accelerates the extraction of CAVs by up to $63.6\times$ (on average $46.4\times$). We provide a theoretical foundation for our approach and give concrete assumptions under which it is equivalent to established SVM-based methods. Our empirical results demonstrate that CAVs calculated with FastCAV maintain similar performance while being more efficient and stable. In downstream applications, i.e., concept-based explanation methods, we show that FastCAV can act as a replacement leading to equivalent insights. Hence, our approach enables previously infeasible investigations of deep models, which we demonstrate by tracking the evolution of concepts during model training.[3]

## 1. Introduction

Humans often think in terms of abstract, perceptually grounded concepts to make sense of the world (Barsalou, 1999). This led to the idea that explaining models in the

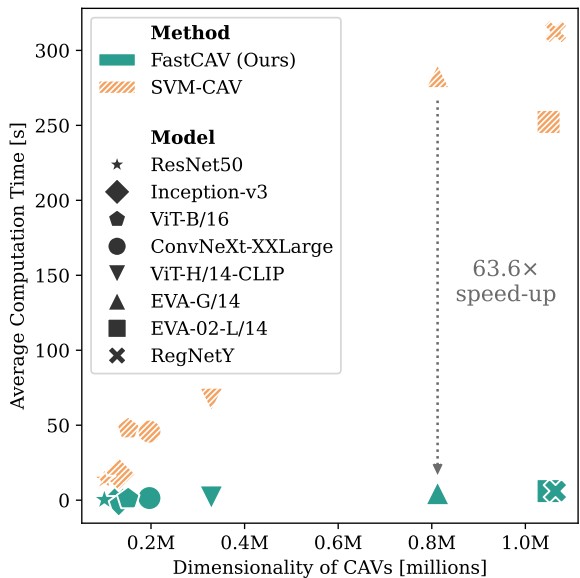

Figure 1: Comparison of computational efficiency between FastCAV and established SVM-CAV for calculating Concept Activation Vectors (CAVs) across different models. The average time to calculate a CAV for each method is plotted against the dimensionality of the activation spaces, demonstrating the significant speedup ($p < 0.01$) achieved by FastCAV.

same conceptual terms makes their behavior more intuitive and relatable to human understanding (Kim et al., 2018; Ghorbani et al., 2019; Yeh et al., 2020; Zhang et al., 2021; Pfau et al., 2021; Nicolson et al., 2024). Concept-based methods investigate the learned representations of a model with respect to human-understandable concepts and quantify the influence of these concepts on the predictions of a selected class. For example: are the concepts "stripes", "wheels", or "green" relevant for the prediction of class *zebra*? To answer such questions, Concept Activation Vectors (CAVs) are introduced in (Kim et al., 2018). CAVs are vectors in the network's activation space that represent human-understandable concepts and are the foundation for many concept-based explainability methods (Ghorbani et al., 2019; Yeh et al., 2020; Oikarinen & Weng, 2024). Specifi-

[1]Institute of Data Science, German Aerospace Center, Jena, Germany [2]Computer Vision Group, Friedrich Schiller University Jena, Germany. Correspondence to: Laines Schmalwasser <Laines.Schmalwasser@dlr.de>.

*Proceedings of the $42^{nd}$ International Conference on Machine Learning*, Vancouver, Canada. PMLR 267, 2025. Copyright 2025 by the author(s).

[3]Project page: https://fastcav.github.io/

cally, these vectors encode directions in a model's activation space, which correspond to changes in a specific semantic concept. For a given class, it is possible to test whether the product of the gradient and vector is positive or negative. If it is positive, the identified concept influences the class. This is called Testing with Concept Activation Vectors (TCAV), (Kim et al., 2018). Following the idea that concepts are encoded as linear directions in the activation space, e.g., (Szegedy, 2013; Karpathy et al., 2015; Goh, 2016; Alain, 2016; Bau et al., 2017; 2020; Elhage et al., 2022), existing works find CAVs by optimizing linear classifiers, usually a linear support vector machine (SVM) (Kim et al., 2018). Further, it is often necessary to calculate multiple CAVs to ensure statistical significance when quantifying the influence on a specific class (Kim et al., 2018).

Unfortunately, these conditions result in long computation times when performing CAV-based analyses. This problem is further exacerbated due to the increasing number of parameters and corresponding activation space dimensionalities in modern architectures. As a concrete example, consider activations after the last convolutional layer of a ResNet50 (He et al., 2016), which has 100.352 dimensions. In comparison, the number of last layer activations in recent models (HuggingFace, 2025), e.g., EVA-02-L/14 (Fang et al., 2024), reaches up to 1.049.600 dimensions. This development and the need for multiple CAV computations result in long processing times and substantial memory consumption. Hence, these constraints can be restrictive for many downstream applications, e.g., when analyzing many layers of a model (Nicolson et al., 2024) or when interpreting numerous concepts simultaneously (Bau et al., 2017). As a result, the practicality of using traditional CAV methods is limited when faced with modern classifiers and tasks. A more efficient approach to compute CAVs in deeper layers without compromising the quality of the insights is needed.

In this work, we propose FastCAV, an approach to speed up the extraction of CAVs within a neural network's activation space by up to $63.6\times$. To achieve this, we follow insights from the area of superposition (Goh, 2016; Elhage et al., 2022) that features in a model are represented as nearly orthogonal vectors in the activation space. Hence, we find the CAV corresponding to a concept by computing the normalized mean vector of the positive concept examples. Additionally, we show that our approach can be understood as an approximation of a Linear Discriminant Analysis (LDA) and, under certain assumptions, it is equivalent to the solution of a linear SVM (Shashua, 1999), providing an intuitive connection to previous works (Kim et al., 2018). We empirically show that, despite the simplicity of our approach, it effectively identifies conceptual directions in high-dimensional activation spaces. Additionally, FastCAV significantly reduces computational expenses, making it more practical for real-world applications where

resources and time are limited. Finally, we apply our approach to track concepts during training and through the layers of a network, which was previously limited by our computational constraints.

The key contributions of our work are as follows:

- We introduce FastCAV, a novel approach for computing Concept Activation Vectors (CAVs) that significantly reduces computation time by up to $63.6\times$, making it more efficient and scalable.

- We provide a theoretical foundation for FastCAV and specify concrete assumptions under which it is equivalent to other linear methods.

- We demonstrate empirically that FastCAV produces similar CAVs to the established SVM-based approach, leading to comparable insights in downstream methods, e.g.,(Kim et al., 2018; Ghorbani et al., 2019).

- Using FastCAV, we investigate the evolution of concepts during training and across different layers of a neural network.

## 2. Related Work

**Concept Activation Vectors (CAVs)**   Alain et al. (Alain, 2016) introduce the idea that features of an intermediate layer are separable using a linear classifier. Based on this, CAVs (Kim et al., 2018) are introduced, which connect abstract model representations and human-understandable concepts. In particular, CAVs represent human-understandable concepts as directions in the activation space of a model's layer. To compute CAVs, a linear classifier is trained to distinguish images that share a visual concept from random images. The CAV corresponding to the concept is proportional to the normal vector of the linear decision boundary. The assumption is that if the concept images cannot be separated in a model's activation space, the model did not adequately learn the concept. Testing with Concept Activation Vectors (TCAV) (Kim et al., 2018) uses CAVs to quantify the sensitivity of a model's predictions for a specific class to the presence of a human-understandable concept. For example, TCAV is able to investigate questions such as whether the color "blue" is relevant for the model when predicting the class *whale*. Further, TCAV and, more specifically, CAVs are the foundation of various concept-based explainability methods, (Ghorbani et al., 2019; Schrouff et al., 2021; Zhang et al., 2021; Singla et al., 2021; Gupta et al., 2024; Ghosh et al., 2023).

Additionally, various methods have been proposed to enhance the interpretability of the discovered concepts. In (Zhang et al., 2021), the authors address directional divergence, and in (Pfau et al., 2021), the authors improve robustness against data variations. Language-guided approaches

like (Moayeri et al., 2023; Huang et al., 2024) remove the dependency on concept images. Instead, they are using vision-text embeddings, e.g., (Radford et al., 2021), and map them to a selected activation space. Hence, they are able to extract CAVs belonging to concepts described by words. In medical imaging, techniques have been developed for bidirectional explanations using regression concept vectors (Graziani et al., 2018). These approaches improved the usability of CAVs, but they remain computationally demanding due to classifier training for each CAV. Further, a recent approach (Pahde et al., 2025) reduces the influence of unrelated distractor patterns, resulting in a similar interpretation of the activation space.

**Linearity in Neural Networks.** Many works study how semantic features or concepts are encoded in neural networks as linear directions in the activation space, e.g., (Szegedy, 2013; Karpathy et al., 2015; Goh, 2016; Alain, 2016; Bau et al., 2017; 2020; Elhage et al., 2022; Bricken et al., 2023; Templeton, 2024; Oikarinen & Weng, 2024). Some identify interpretable neurons (Karpathy et al., 2015; Bau et al., 2017; 2020; Oikarinen & Weng, 2022; 2024). In contrast, other studies suggest that neurons can also be polysemantic (Olah et al., 2017; 2020), meaning that a neuron responds to more than one feature or concept. In (Elhage et al., 2022), the authors find that a network learns to encode features as a combination of neurons, i.e., a direction in feature space that does not align with the axes. They argue that this happens if there are more features than neurons because features cannot be equally distributed across the neurons in that case.

Based on this idea, different works explore the phenomenon that a neural network can encode more features than dimensions in a hidden representation, (Olah et al., 2020; Elhage et al., 2022; Bricken et al., 2023; Templeton, 2024). The strategy to achieve this is called *superposition* (Olah et al., 2020). Elhage et al. (Elhage et al., 2022) show on a synthetic dataset that a neural network in *superposition* encodes the features as almost-orthogonal directions to each other. Recent work explores superposition further for large language models (Bricken et al., 2023; Templeton, 2024).

In this work, we follow the idea that features are encoded as almost orthogonal linear directions in activation space to design FastCAV.

# 3. Methodology

## 3.1. Preliminaries

Concept-based explanation approaches aim to explain trained neural networks $f$ on the level of human-understandable concepts. Specifically, they investigate the activation space after a layer $l$. Hence, we split $f$ into the

concatenation of two mappings: $g_l$, which comprises the first $l$ layers, and $h_l$, which consists of the remaining layers. For an input $x$, we write $f(x) = h_l(g_l(x))$.

To interpret $g_l(\cdot)$ with respect to a semantic concept $c$, we follow (Kim et al., 2018) and describe a concept by a set of user-defined images $D_c$. This visual definition allows for a specification independent of the particular training task and data. The model $f$ has learned concept $c$ after layer $l$ if it is possible to separate images of $D_c$ from random samples of a set $D_r$ in the respective activation space. Specifically, in (Kim et al., 2018), the authors learn linear SVM classifiers to identify linear combinations of activations that correspond to a concept $c$. They define a Concept Activation Vector (CAV) $v_c^l$ as the normal vector of such a learned linear decision boundary. For binary labels $\text{sgn}(y)$ describing the membership of $x$ in $D_c$ or $D_r$, the linear classifier is

$$y = v_c^l \cdot g_l(x) + b, \tag{1}$$

where $b$ is the intersect, and $\cdot$ is the dot-product.

Intuitively, $v_c^l$ can be understood as the direction in the activation space of $g_l$ that corresponds to a concept $c$. If the resulting classifier defined in Equation (1) achieves a low separation performance, then $f$ has not sufficiently learned concept $c$ (Kim et al., 2018). In the following section, we detail our alternative approach to calculating CAVs.

## 3.2. FastCAV

Following insights from previous works on orthogonality in feature space, e.g., (Olah et al., 2020; Elhage et al., 2022), FastCAV aims to identify the direction directly as the difference between random images and a concept set. Specifically, to find a CAV $v_c^l$ for a set of concept images $D_c$, we first calculate the mean of $g_l(x)$ for all $x \in D_c \cup D_r$, i.e., we use a maximum likelihood estimator, (Bishop, 2006),

$$\hat{\mu}_{D_c \cup D_r} = \frac{1}{|D_c| + |D_r|} \sum_{x \in D_c \cup D_r} g_l(x). \tag{2}$$

We now use this global mean to normalize all activation vectors $g_l(x)$ for samples in $D_c$. Then, we follow the intuition detailed above that $v_c^l$ encodes the direction in activation space that separates samples contained in $D_c$ from samples in $D_r$. Specifically, we calculate

$$v_c^l \propto \frac{1}{|D_c|} \sum_{x \in D_c} (g_l(x) - \hat{\mu}_{D_c \cup D_r}), \tag{3}$$

which we normalize to unit length, following the SVM approach (Kim et al., 2018).

Intuitively, we first center all activations and then find the direction pointing towards the concept examples. Hence,

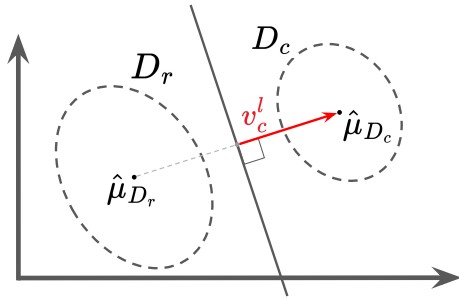

Figure 2: Schematic illustration of FastCAV in two dimensions. Specifically, the learned concept activation vector $v_c^l$ points from the global mean $\hat{\mu}_{D_c \cup D_r}$ towards $\hat{\mu}_{D_c}$.

in FastCAV, $v_c^l$ is the vector which points from $\hat{\mu}_{D_c \cup D_r}$ towards the mean of $g_l(x)$ for all $x \in D_c$, see Figure 2.

With Equation (1) and $y = 0$, the intersect $b$ is given by

$$b = -v_c^l \cdot \hat{\mu}_{D_c \cup D_r}, \qquad (4)$$

fully specifying the linear decision boundary of FastCAV.

### 3.3. Connection to Other Linear Classifiers

We study FastCAV on a theoretical level to investigate the connection to the established linear SVM approach for finding CAVs, e.g., (Kim et al., 2018). In detail, we show that under certain assumptions, both approaches identify the same direction in latent space. While these assumptions are strong, they provide an intuition for the strong performance of our approach in our empirical evaluations.

In Section 3.2, we describe FastCAV in terms of concrete observations $x \in D_c \cup D_r$. These observations are generated by some process and follow the latent distribution of inputs. Hence, we can study this process in terms of random variables following this unknown distribution. Further, $g_l$ of the trained model $f$ leads to a mixture distribution for random images and images conditioned on the concept $c$. In other words, we cast the separation of random and concept samples as a statistical learning problem, (Vapnik, 1998).

Specifically, we consider the maximum likelihood estimate of the global mean in Equation (2). This estimator is generally unbiased under *i.i.d.* samples following a Gaussian distribution (Bishop, 2006). Further, in Appendix A.1, we show that if both the random samples and concept samples follow a multivariate Gaussian distribution and are equally mixed, then the expectation of the estimator is

$$\mathbb{E}[\hat{\mu}_{D_c \cup D_r}] = \frac{\mu_c + \mu_r}{2}, \qquad (5)$$

where $\mu_c$ and $\mu_r$ are the respective means of the Gaussians.

Then FastCAV finds $v_c^l$ as the unit vector proportional to the vector pointing from the global mean to the mean of

the concept examples $\mu_c$. In Equation (3), we again apply the respective maximum likelihood estimator $\hat{\mu}_{D_c}$, meaning $v_c^l \propto \hat{\mu}_{D_c} - \hat{\mu}_{D_c \cup D_r}$. Under the assumptions specified above and again drawing an expectation over the sample of random inputs and concept examples, we get

$$\mathbb{E}[v_c^l] \propto \mu_c - \frac{\mu_c + \mu_r}{2} = \frac{\mu_c - \mu_r}{2}. \qquad (6)$$

This solution is a simplified form of linear discriminant analysis (LDA), more specifically, Fisher discriminant analysis (Bishop, 2006) under precise assumptions. Using a related maximum likelihood estimator for the total within-class covariance

$$\hat{\Sigma}_{D_c \cup D_r} = \sum_{x \in D_c} (g_l(x) - \hat{\mu}_c)(g_l(x) - \hat{\mu}_c)^\top \\ + \sum_{x \in D_r} (g_l(x) - \hat{\mu}_r)(g_l(x) - \hat{\mu}_c), \qquad (7)$$

the Fisher discriminant solution for the normal vector of the linear decision boundary between the two classes is proportional to $\hat{\Sigma}_{D_c \cup D_r}^{-1}(\hat{\mu}_c - \hat{\mu}_r)$. Under the explicit additional assumption of isotropic within-class covariances, i.e., $\Sigma^{-1}$ being proportional to the unit matrix, the solution for the normal vector is equivalent to FastCAV. To validate this approach, we empirically compare against the full Fisher LDA in Appendix B.2.1 and find faster computation times and higher accuracies for various model architectures when using FastCAV.

To summarize, under the assumptions of isotropic within-class covariances and Gaussian distributions for both random and concept examples, FastCAV is equivalent to the solution of Fisher discriminant analysis on corresponding activations. Further, in (Shashua, 1999), the authors investigate the relationship between a linear Fisher discriminant and a linear SVM, which is the established classifier utilized in (Kim et al., 2018) to compute CAVs. Specifically, they find that the solution of Fisher discriminant analysis on the set of support vectors is equivalent to the solution of the linear SVM. Hence, a support vector machine can be understood as a sparsification of the linear Fisher discriminant.

Finally, consider that the aim is to explain the learned representation of a model after layer $l$. Hence, these activation vectors $g_l(x)$ are potentially of a very high dimensionality $d$. Further, let $n = |D_c \cup D_r|$ be the number of samples and assume $d >> n$. Then the set of support vectors likely contains many of the samples in $D_c \cup D_r$. Related works study this in more depth, e.g., (Muthukumar et al., 2021; Hsu et al., 2021), and find that the fraction of support vectors increases with increasing dimensionality. We empirically confirm this and find high ratios of support vectors when studying the activations of modern architectures with linear SVM classifiers (see Table 3 in Appendix A.2).

To conclude, under the assumptions detailed above and in high-dimensional activation spaces, the normal vectors of the hyperplanes found by FastCAV and a linear SVM used in (Kim et al., 2018) are identical. Further, we find that in practice, our approach leads to similar solutions while being much faster to calculate, which we demonstrate in our experimental investigations.

### 3.4. Complexity Analysis

The overall time complexity for training an SVM is $\mathcal{O}(\max(n, d) \min(n, d)^2)$, where $n$ is the number of samples and $d$ is the dimensionality of the input space. This holds for both the primal and the dual optimization problem, (Chapelle, 2007). However, the complexity and speed also depend on the selected kernel, which is linear in our case (Kim et al., 2018). Hence, to benchmark the speed, we utilize various common implementations of linear SVMs. First, to solve the dual problem, we use the Sequential Minimal Optimization (SMO) algorithm (Platt, 1998). Specifically, we use the implementation in (Pedregosa et al., 2011), which uses the SMO implementation from (Chang & Lin, 2011). Second, regarding the primal problem, we use the implementation in (Fan et al., 2008), which is also included in (Pedregosa et al., 2011). Lastly, using Hinge loss, it is possible to approximate a linear SVM classifier using stochastic gradient descent (SGD), e.g., (Pedregosa et al., 2011). For this last implementation specifically, the complexity boils down to calculating a gradient with respect to the linear decision boundary parameters ($\mathcal{O}(d)$) for each of the $n$ training samples for $T$ iterations. Hence, the SGD variant in (Pedregosa et al., 2011) has a complexity of $\mathcal{O}(Tnd)$. We benchmark these three implementations and provide the results in Appendix A.3. In our use case, we find that SGD-based optimization performs best.

In comparison, FastCAV calculates a sum of $n$ vectors with dimension $d$ and performs $d$ multiplications. Hence, training complexity is $\mathcal{O}(nd)$ with very small constants. Once the linear decision boundary is found, the inference for both SVM-CAV and FastCAV is equivalent to the dot product between two $d$-dimensional vectors, resulting in $\mathcal{O}(d)$.

## 4. Experiments

With FastCAV, we propose an alternative to calculate CAVs, which is more computationally efficient. To demonstrate its effectiveness, we compare our approach against the established SVM-based approach ("SVM-CAV") (Kim et al., 2018) in terms of speed and CAV quality. We repeat this comparison for more specialized medical concepts to highlight the broad applicability. Further, a comparison to other CAV calculation methods is provided in Appendix B.2.1, and a discussion of failure cases is provided in Appendix B.2.2. Next, we show that FastCAV can work

as a replacement in downstream concept-based explanation methods and generate similar insights. In particular, we compare FastCAV against SVM-CAV and investigate Testing with Concepts Activation Vectors (TCAV) (Kim et al., 2018), and Automatic Concept-based Explanations (ACE) (Ghorbani et al., 2019). Finally, we utilize the increased efficiency of FastCAV to track CAVs throughout the training and layers of a ResNet50 (He et al., 2016) to understand how concepts evolve. We provide the complete technical setup and comprehensive details for all our experiments in Appendix B.1.

### 4.1. FastCAV versus SVM-based CAV computation

To empirically compare FastCAV and SVM-CAV, we evaluate a broad spectrum of architectures trained on ImageNet (Russakovsky et al., 2015). Additionally, we investigate run times for the CAV computation in a realistic setting and select current top-performing model architectures from the `timm`-leaderboard (Wightman, 2019). We used the models reported in the first column of Table 1, i.e., (Szegedy et al., 2016; He et al., 2016; Liu et al., 2022; Singh et al., 2022; Dosovitskiy et al., 2021; Fang et al., 2023; 2024).

In our setup, we sample 60 concept images per set $D_c$ from the Broden dataset (Bau et al., 2017). Similarly, we sample 60 random images from ImageNet (Russakovsky et al., 2015) for each set $D_r$. All our results are reported as averages over 30 resampled $D_r$ across all concepts of the Broden dataset, (Bau et al., 2017), and over the respective network layers. We split our investigation into four dimensions: computational time, accuracy, inter-method similarity, and intra-method robustness.

**Computational time** We analyze the theoretical time complexity of FastCAV and SVM-CAV in Section 3.4. As a complementary analysis, we now compare the run time of both methods in practice. In Table 1, columns "Comput. time [$s$]" report the respective average times to compute one CAV per model. In Figure 1, we visualize the timings for the pre-final network layers before classification. A similar visualization for the average over all layers is included in Appendix B.2.

Both methods require more time to calculate CAVs for models with higher activation space dimensions. However, this observation is expected and follows from the complexity of both methods, which scale linearly in the number of dimensions $d$. Nevertheless, under practical conditions, FastCAV is up to $63.6\times$ (on average $46.4\times$) faster than SVM-CAV.

**Accuracies of CAVs to distinguish between $D_c$ and $D_r$** In addition to the increased computation speed of FastCAV, we need to evaluate the quality of the discovered CAVs. Hence, we compare the predictive performance, i.e., accu-

Table 1: Comparing our approach FastCAV with SVM-CAV. **Bold values** indicate better results. "N/A" indicates that no results were produced due to the overall computational time for the respective model exceeding four days. Additional results for other CAV calculation methods can be found in Appendix B.2.1.

| Model | ∅ Dim. | Comp. Time [$s$] ↓ | | Accuracy ↑ | | Similarity ↑ | | |
|---|---|---|---|---|---|---|---|---|
| | | FastCAV | SVM-CAV | FastCAV | SVM-CAV | FastCAV | SVM-CAV | Inter-Method |
| Inception-v3 | 206,169 | **0.4**±**0.14** | 44.7±29.3 | **0.95**±**0.06** | 0.93±0.09 | **0.795**±**0.013** | 0.338±0.062 | 0.898±0.053 |
| ResNet50 | 341,197 | **1.1**±**1.51** | 135.4±112.9 | **0.89**±**0.15** | 0.87±0.15 | **0.816**±**0.029** | 0.400±0.062 | 0.837±0.146 |
| ConvNeXt-XXL | 753,830 | **5.5**±**3.88** | 269.6±167.0 | 0.90±0.08 | **0.93**±**0.07** | **0.831**±**0.042** | 0.449±0.075 | 0.807±0.136 |
| RegNetY | 1,869,479 | **8.2**±**1.04** | N/A | 0.90±0.05 | N/A | 0.734±0.009 | N/A | N/A |
| ViT-B/16 | 140,560 | **1.1**±**0.03** | 50.5±9.2 | **0.82**±**0.13** | 0.81±0.14 | **0.787**±**0.024** | 0.408±0.046 | 0.818±0.064 |
| ViT-H/14-CLIP | 281,925 | **1.9**±**0.08** | 69.0±10.9 | **0.87**±**0.15** | 0.86±0.15 | **0.603**±**0.095** | 0.249±0.090 | 0.858±0.083 |
| EVA-G/14 | 696,298 | **4.7**±**0.14** | 248.9±60.3 | **0.88**±**0.14** | 0.86±0.14 | **0.650**±**0.082** | 0.250±0.080 | 0.888±0.060 |
| EVA-02-L/14 | 899,653 | **6.1**±**0.25** | 301.6±114.7 | **0.89**±**0.14** | 0.88±0.16 | **0.675**±**0.077** | 0.219±0.076 | 0.836±0.095 |

racy, of the linear classifier posed by Equation (4) using CAVs obtained from our approach and SVM-CAV.

The results are presented in Table 1 ("Accuracy"). Notably, FastCAV results in slightly higher accuracies, with one exception (ConvNeXt-XXL). Nevertheless, both methods yield similar results. We conclude that FastCAV does not compromise the predictive quality of discovered CAVs.

**Inter-method similarity**   To investigate the agreement between the generated CAVs of both methods, we directly compare the computed CAVs. For that, we report in Table 1 ("Inter-Method") the average cosine similarity between CAVs calculated by FastCAV and by SVM-CAV.

Across all models, we observe high similarities (values between 0.8 and 0.9). This result indicates that both FastCAV and SVM-CAV find similar directions in diverse activation spaces for a variety of concepts.

**Intra-method robustness**   Lastly, we evaluate the variation of the discovered CAVs across different $D_r$ for both methods. In Table 1, the first two columns of "Similarity" report the average pairwise cosine similarity of the set of 30 CAVs aggregated across all concepts and network layers.

Overall, we find FastCAV computes CAVs of a higher pairwise similarity. This increased robustness of FastCAV is due to the CAVs only differing because of the estimations of the global mean $\hat{\mu}_{D_c \cup D_r}$ while $D_c$ stays fixed. In contrast, SVM-CAV completely relearns linear SVMs using a non-deterministic SGD implementation (see Section 3.4).

## 4.2. Medical Task

While our main experiments focus on a general-purpose model and task, we show that FastCAV can also be applied in more specialized scenarios. Following prior work on medical concept extraction (Singla et al., 2021), we fine-tune a DenseNet-121 (Huang et al., 2017) on a curated subset of the

Table 2: Comparing FastCAV with SVM (Ghosh et al., 2023) and sparse logistic regression (SL-R) (Singla et al., 2021) for specialized medical concepts in a chest x-ray image classification task. **Bold values** indicate better results.

| Method | C. Time [$s$] ↓ | Accuracy ↑ | Similarity ↑ |
|---|---|---|---|
| FastCAV | **0.006**±**0.001** | 0.72±0.11 | **0.79**±**0.03** |
| SL-R | 6.370±0.756 | **0.72**±**0.13** | 0.50±0.04 |
| SVM | 0.439±0.071 | 0.71±0.13 | 0.46±0.04 |

MIMIC-CXR dataset (Johnson et al., 2019). Next, we use the CheXpert labeler (Irvin et al., 2019) to extract concept mentions from the included free-text radiology reports included with MIMIC-CXR (Johnson et al., 2019). Using the resulting concepts, we derive CAVs for each of the model's four dense blocks applying FastCAV, SVM-based computation (Ghosh et al., 2023), and sparse logistic regression (Singla et al., 2021). In Table 2, we report the average computation time, CAV accuracy, and intra-method similarity (see Section 4.1). FastCAV achieves performance similar to that of the established approaches in this domain. However, note the strongly reduced computation time and improved similarity. We include an evaluation of individual concepts and the technical details of our setup in Appendix B.2.3.

## 4.3. CAV-based Explanation Methods

After quantitatively comparing CAVs computed by our approach and the SVM-based approach, we investigate the qualitative performance with downstream concept-based explanation methods. For this comparison, we substitute SVM-based CAV computation with FastCAV and contrast the generated explanations, showcasing its ability to act as a drop-in replacement. As concept-based explanation methods, we select Testing with Concept Activation Vectors (TCAV) (Kim et al., 2018) and Automatic Concept-based Explanations (ACE) (Ghorbani et al., 2019).

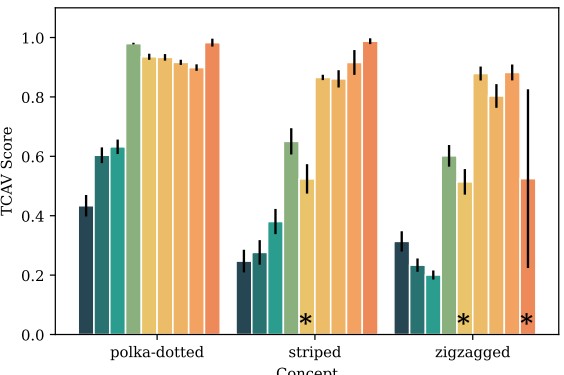

Figure 3: TCAV scores for various GoogleNet (Szegedy et al., 2015) layers. We compare the concepts "polka-dotted", "striped", and "zigzagged" for the class *ladybug* using FastCAV against SVM-CAV. We follow (Kim et al., 2018) and mark CAVs that are not statistically significant with "*".

**Testing with Concept Activation Vectors (TCAV)**
TCAV calculates the directional derivative of a selected class $k$ with respect to the activations of a layer $l$, i.e., $g_l(x)$. For a given input $x$, the sensitivity score for a concept $c$ with corresponding CAV $v_c^l$ is $\nabla h_{l,k}(g_l(x)) \cdot v_c^l$. The corresponding TCAV score is now the fraction of a set of inputs of the selected class that have positive sensitivity scores. Intuitively, this scalar measures the importance of a concept in layer $l$ for the output node corresponding to class $k$.

We qualitatively evaluate FastCAV by repeating an experiment of (Kim et al., 2018). Specifically, we select all layers of GoogleNet (Szegedy et al., 2015) trained on ImageNet (Russakovsky et al., 2015) and calculate the TCAV scores for various classes and concepts. In Figure 3, we compare the results for FastCAV against SVM-CAV for the class *ladybug* and the concepts "polka dotted", "striped", and "zigzagged". In Appendix B.3, we include more examples.

The insights into the GoogleNet model are consistent between both our approach and the SVM-based method. For *ladybug*, the concepts "polka dotted" and "striped" are more important than "zigzagged". This observation especially applies in later layers of the network. Further, both methods agree that "polka dotted" is the most important for earlier layers. We hypothesize that the usage of "striped" in later layers is due to blades of grass in many *ladybug* images.

In contrast to the similar average results, we observe a clear distinction for the standard deviations of the TCAV score over multiple repetitions. Here, FastCAV leads in most cases to a smaller variance in the measured scores. This observation confirms our intra-method robustness results in Table 1. Additionally, this result and the overall similarity between both FastCAV and SVM-CAV hold for other qualitative examples in Appendix B.3. Hence, we conclude that our approach leads to similar model explanations using TCAV as SVM-based CAV computation.

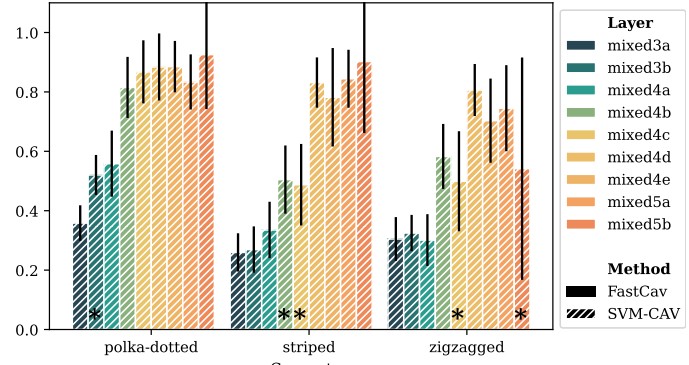

(a) FastCAV       (b) SVM-CAV

Figure 4: Most salient concepts discovered by ACE (Ghorbani et al., 2019) using either our FastCAV or the established SVM-CAV. In both cases, we find the discovered patches containing stripes, which is congruent with the original observation in (Ghorbani et al., 2019).

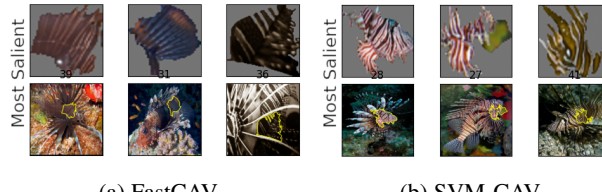

(a) Impact of $|D_c|$.       (b) Number of resampled $D_r$.

Figure 5: Sensitivity analysis of FastCAV to (a) the number of concept images and (b) number of resampled $D_r$. Note the differences in y-axis scales.

**Automatic Concept-based Explanations (ACE)** ACE is based on TCAV and proposes to automatically discover and extract concepts that are important for a selected class. In detail, they perform superpixel segmentation on a set of images from class $k$ using multiple resolutions to capture hierarchical concepts. Next, these superpixels are grouped into concept sets using (Zhang et al., 2018). The resulting sets are pruned and then ordered according to their importance for class $k$ using TCAV scores.

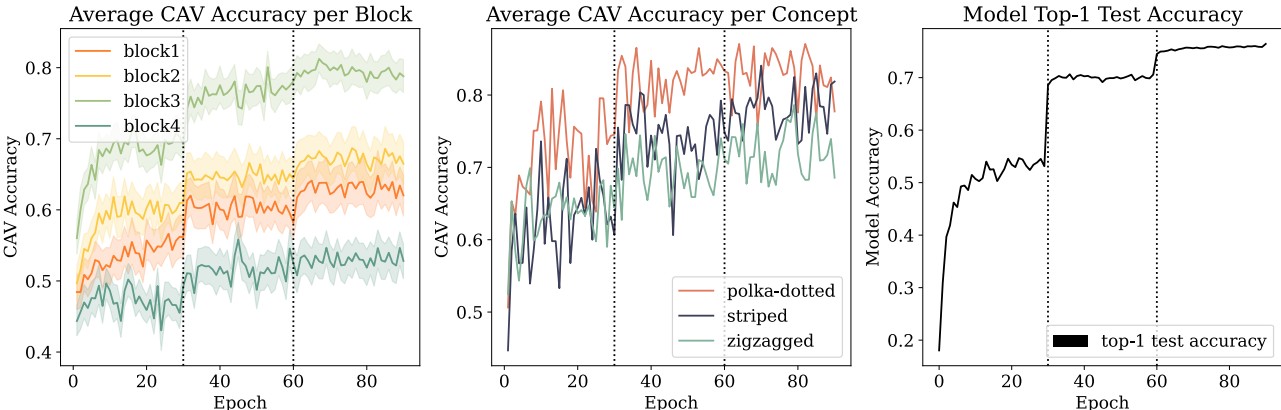

Figure 6: Evolution of various CAVs during training of a ResNet50 (He et al., 2016) on ImageNet (Russakovsky et al., 2015). On the left, we visualize the average accuracies achieved by CAVs after the final layers in each of the four ResNet blocks. In the middle, we investigate CAVs for three specific concepts. On the right, we display the test accuracy during training. The vertical lines indicate epochs, after which the learning rate is divided by ten. Note the differences in y-axis scales.

To evaluate FastCAV for ACE qualitatively, we follow (Ghorbani et al., 2019) by analyzing the mixed_7a layer in an Inception-v3 model (Szegedy et al., 2016) trained on ImageNet (Russakovsky et al., 2015).

Figure 4 visualizes the most salient concepts discovered with ACE. We include additional examples in Appendix B.4, following the class selection in (Ghorbani et al., 2019). In all cases, we find similar example patches for the discovered concepts. In particular, the concept "stripes" is important for the class *lionfish* irrespective of whether we employ Fast-CAV or SVM-CAV. Further, the TCAV scores for the most salient concepts are similar, with 0.78 for our approach and 0.73 for SVM-CAV. These similarities between the discovered concepts and scores also apply to the other examples in Appendix B.4. Hence, we conclude that FastCAV is suitable for automatic concept discovery using ACE.

### 4.4. FastCAV Sensitivity Analysis

Our approach to finding CAVs depends on the number of concept images, i.e., $|D_c|$ and $|D_r|$. Hence, it is important to understand how many samples are necessary to compute robust concept directions. Additionally, the robustness also depends on the number of random sets $D_r$ which are used for statistical testing, (Kim et al., 2018).

Here, we conduct a sensitivity analysis to explore the influence of both variables. In particular, we vary the sizes of $D_c$ and $D_r$ as well as the number of available random sets. Further, for this sensitivity analysis, we compute CAVs using FastCAV for various networks. Specifically, we use the pre-final layers of ResNet50 (He et al., 2016), ViT-16, (Dosovitskiy et al., 2021), and Inception-v3 (Szegedy et al., 2016) pretrained on ImageNet (Russakovsky et al., 2015).

**Influence of** $|D_c| = |D_r|$ We fix the total number of available random sets to 60 and vary the number of examples in the concept and random sets equally, evaluating the corresponding CAV accuracies. In Figure 5a, we visualize the influence of $|D_c|$ on the accuracy of the CAVs to distinguish between concept and random images. In particular, we find that for all three models, the performance remains high with more than 60 samples. This indicates that with at least 60 examples, the CAV training process effectively captures the defining features of the concept. Following this observation, we recommend using at least 60 examples per concept.

**Influence of the Number of Random Sets** $D_r$ Following the previous sensitivity analysis, we fix the number of examples in $D_c$ and $D_r$ to 60 and vary the number of random sets. In Figure 5b, we visualize the corresponding sensitivity of the CAV accuracies. We see expected stabilization, i.e., lower variances, when using more random sets $D_r$. However, the average accuracy remains approximately constant. Hence, in our final experiment, we select 100 random sets.

### 4.5. Tracking CAVs During Training

**Setup** Using FastCAV, we are able to track concepts during the training of a model with a high number of parameters. To show this, we train a ResNet50 (He et al., 2016) on the ImageNet dataset (Russakovsky et al., 2015) from scratch. We follow the same setup as (Paszke et al., 2019) and include the full details in Appendix B.5. As concepts, we select the textures from (Bau et al., 2017). Examples include "polka-dotted", "striped", and "zigzagged". Specifically, we compute CAVs against 100 random sets after each training epoch and select the final layers of each of the four ResNet blocks. Our analysis focuses on when and in which layer the model learns concepts during the training.

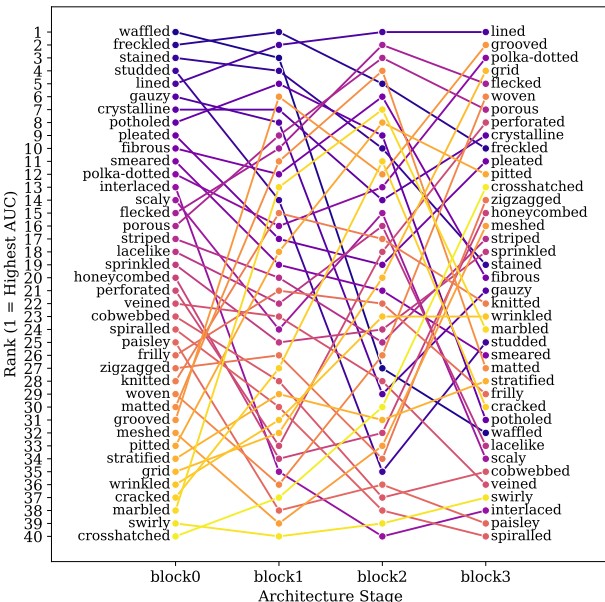

Figure 7: Evolution of various CAVs during training of a ResNet50 (He et al., 2016) on ImageNet (Russakovsky et al., 2015). We report the concept ranking of the achieved AUC scores during training for the different network blocks.

**Results** Figure 6 demonstrates that the model learns concepts during training, with visibly increasing CAV accuracy after each step. This aligns with improved model accuracy, suggesting that the model learns to employ relevant concepts for predictions. We observe similar trends for specific concept examples, although results exhibit increased variability across the training steps compared to the average across concepts. These results indicate that the performance of relevant CAVs correlates with overall model performance.

Notably, we observe stark increases in average CAV accuracy after each epoch, where the learning rate is divided by ten during training. This observation aligns with abrupt increases in the top-1 test accuracy. Furthermore, we find that early and middle layers have a higher likelihood of learning textures than later layers. This result validates findings in other works like (Kim et al., 2018; Ghorbani et al., 2019; Bau et al., 2017). Our observations provide further evidence for this hypothesis and demonstrate that our approach can be used to study the learning dynamics of deep neural networks in a more fine-grained manner and for abstract concepts.

Lastly, we conduct an analysis ranking concepts by their area under the curve (AUC) during training and across layers. Specifically, high AUC values indicate that a concept in a certain layer is learned early, and the CAV achieves sufficient accuracy during the complete training. We visualize the resulting rankings over the network blocks in Figure 7 and observe notable changes in the learnability

of concepts across layers. For instance, concepts that rank highly in earlier network blocks may be less important and rank lower in later ones. As a concrete example, consider the concept "wave", which is the highest-ranked concept in the first block. However, its ranking drops to second after block two and ultimately deteriorates to 32nd in the final block. In contrast, the concept "lined" remains among the highest-ranked concepts throughout all layers. This observation underscores how the model prioritizes different concepts at various stages, with some foundational concepts giving way to more refined or abstract ones in deeper layers.

## 5. Conclusion

In this work, we introduced FastCAV, a novel approach to efficiently compute Concept Activation Vectors (CAVs) for explaining deep neural networks. By leveraging ideas from superposition (Elhage et al., 2022), our method accelerates CAV extraction. Specifically, we take the normalized vector pointing towards the concept example mean in activation space, which is significantly faster than learning a linear SVM classifier. We further provide a theoretical foundation for our method and give specific assumptions under which FastCAV is equivalent to the established SVM-based approach. Our empirical results demonstrate that FastCAV achieves a significant reduction in computational cost, with speedups of up to $63.6\times$ (and an average speedup of $46.4\times$), while maintaining at least similar CAV quality. We further investigate the variations in downstream performances using TCAV (Kim et al., 2018) and ACE (Ghorbani et al., 2019), and find only marginal differences in the results. To showcase the capabilities of FastCAV, we apply it to analyze the training of a model, tracking the evolution of concepts during training and across network layers. Our analysis validates the notion that textural concepts are learned in the middle layers of a network, aligning with prior works (Bau et al., 2017; Kim et al., 2018; Ghorbani et al., 2019).

Overall, FastCAV serves as a drop-in replacement to calculate CAVs, which leads to lower run times and maintains high quality. While we thoroughly evaluate the performance of the generated CAVs, we acknowledge that further analysis of other CAV properties, such as locality, consistency, and entanglement(Nicolson et al., 2024), is an interesting direction for future work. We hope that the gain in computational speed of FastCAV facilitates concept-based explanations that were previously infeasible.

## Impact Statement

This paper presents work whose goal is to advance the field of Machine Learning, specifically, explainability. There are many potential societal consequences of our work, none of which we feel must be specifically highlighted here.

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

# A. FastCAV— Additional Details

In this section, we provide additional theoretical details connected to Section 3 in our main paper.

## A.1. Expectation of the Global Mean Under Gaussian Mixture Assumption

In FastCAV, we utilize the maximum likelihood estimator for the mean of a sample. This estimator is generally unbiased under *i.i.d.* samples following a Gaussian distribution, e.g., (Bishop, 2006). In Equation (2) in our main paper, we apply this to the global mean over $D_c \cup D_r$. However, consider the following related estimators:

$$\hat{\mu}_{D_c} = \frac{1}{|D_c|} \sum_{x \in D_c} g_l(x), \quad \text{and} \quad \hat{\mu}_{D_r} = \frac{1}{|D_r|} \sum_{x \in D_r} g_l(x). \tag{8}$$

We now assume that both $D_c$ and $D_r$ contain *i.i.d.* samples so that the corresponding activations $g_l(x)$ follow two respective Gaussian distributions $\mathcal{N}(\mu_c, \Sigma_c)$ and $\mathcal{N}(\mu_r, \Sigma_r)$. In other words, we assume that both the random sample and concept example activations follow independent Gaussian distributions in the activation space of $g_l$. Then from the unbiasedness of the maximum likelihood estimator (Bishop, 2006) follows both

$$\mathbb{E}[\hat{\mu}_{D_c}] = \mu_c, \quad \text{and} \quad \mathbb{E}[\hat{\mu}_{D_r}] = \mu_r. \tag{9}$$

Using these equations, we now study the expectation of the global mean estimator $\hat{\mu}_{D_c \cup D_r}$. In (Kim et al., 2018), both the random sample and the set of concept examples are of equal size, which we follow in our work. Hence, we assume a uniform mixture of the two Gaussians specified above. In other words, the mixture weights are $1/2$, and $|D_c| = |D_r|$. Then the expectation of Equation (2) becomes

$$\mathbb{E}[\hat{\mu}_{D_c \cup D_r}] = \mathbb{E}\left[ \frac{1}{|D_c \cup D_r|} \sum_{x \in D_c \cup D_r} g_l(x) \right] \tag{10}$$

$$= \mathbb{E}\left[ \frac{1}{|D_c| + |D_r|} \sum_{x \in D_c} g_l(x) + \frac{1}{|D_c| + |D_r|} \sum_{x \in D_r} g_l(x) \right] \tag{11}$$

$$= \mathbb{E}\left[ \frac{1}{2} \frac{1}{|D_c|} \sum_{x \in D_c} g_l(x) + \frac{1}{2} \frac{1}{|D_r|} \sum_{x \in D_r} g_l(x) \right] \tag{12}$$

$$= \frac{1}{2} \left( \mathbb{E}\left[ \frac{1}{|D_c|} \sum_{x \in D_c} g_l(x) \right] + \mathbb{E}\left[ \frac{1}{|D_r|} \sum_{x \in D_r} g_l(x) \right] \right) \tag{13}$$

$$= \frac{1}{2} \left( \mathbb{E}[\hat{\mu}_{D_c}] + \mathbb{E}[\hat{\mu}_{D_r}] \right) \tag{14}$$

$$= \frac{\mu_c + \mu_r}{2}. \tag{15}$$

Intuitively, the global mean is the middle point between the mean of the random samples and the concept examples in the activation space of $g_l$. We then use this point to normalize the activations before calculating the direction towards the concept mean as our CAV $v_c^l$. This vector is under the specified assumptions and an expectation over the sample $D_c \cup D_r$ proportional to Equation (6) in our main paper.

## A.2. Empirical Ratio of Support Vectors

Table 3: Percentage of support vectors of SVMs that learned the representation of 40 different concepts (Bau et al., 2017). Each SVM was trained to separate 50 example concept images from 50 random images. We average over 30 redrawn $D_r$ and split our report into the two classes defined by the sets $D_c$ and $D_r$.

| | | Support Vector Percentage | |
|---|---|---|---|
| Model Architecture | ∅ Activation Dimensionality | $D_c$ | $D_r$ |
| ResNet50 (He et al., 2016) | 100,352 | 77.05% | 98.17% |
| ConvNeXt-XXL (Liu et al., 2022) | 196,608 | 52.31% | 90.40% |
| RegNetY (Singh et al., 2022) | 1,064,448 | 78.00% | 97.66% |
| ViT-B/16 (Dosovitskiy et al., 2021) | 151,296 | 73.27% | 78.87% |
| ViT-H/14-CLIP (Dosovitskiy et al., 2021) | 328,960 | 87.85% | 88.72% |
| EVA-G/14 (Fang et al., 2023) | 812,416 | 78.69% | 90.60% |
| EVA-02-L/14 (Fang et al., 2024) | 1,049,600 | 88.45% | 99.67% |

In Table 3, we list the percentages of support vectors when using an SVM to compute CAVs in the pre-final layer of various networks. In all cases, we observe that the majority of activation vectors are support vectors. This observation is congruent with other works that study this behavior for high dimensionalities in more detail, e.g., (Muthukumar et al., 2021; Hsu et al., 2021). However, interestingly, we find that the percentages are higher for the sets of random images $D_r$ in comparison to concept images $D_c$. We hypothesize that this might be a consequence of more compact activations for images sharing a semantic concept.

## A.3. Runtime Benchmark of SVM Implementations

As described in our main paper, we benchmark the speed of three common implementations of linear SVMs. First, the SMO (Platt, 1998) implementation contained in (Pedregosa et al., 2011), which relies on (Chang & Lin, 2011). Second, we use the solution for the primal problem implemented in (Fan et al., 2008), which is also included in (Pedregosa et al., 2011). Lastly, using Hinge loss, it is possible to approximate a linear SVM classifier using stochastic gradient descent (SGD), e.g., (Pedregosa et al., 2011). We visualize the measured achieved runtimes in Figure 10 for both increasing numbers of examples $n$ and increasing input dimensions $d$. For our application, we find that the SGD-based optimization performs best. Hence,

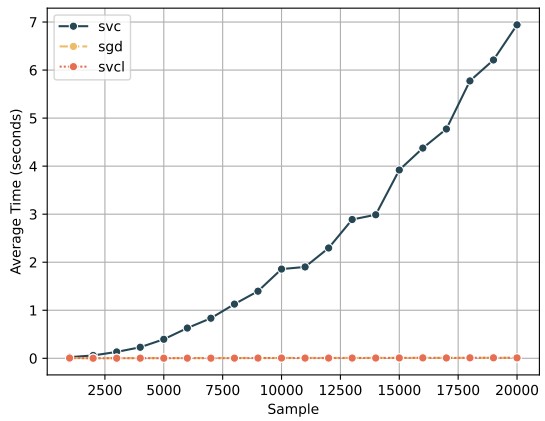 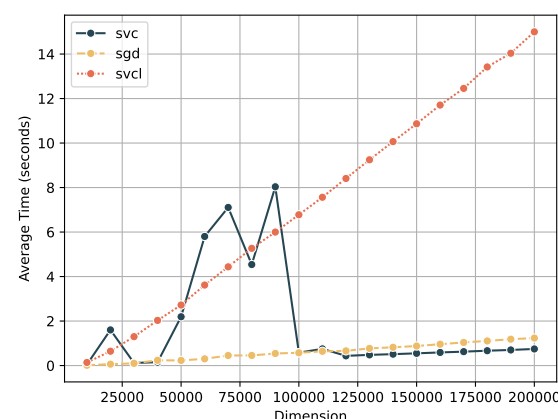

(a) Dependency on the number of samples $n$. We fix the dimensionality to $d = 2$.

(b) Dependency on dimensionality $d$. We set the total number of samples to $n = 50$.

Figure 8: It can be seen that our method results in similar results.

we use this implementation for the classical SVM-based CAV calculation in our experiments. Note that this also follows

related work (Kim et al., 2018), which uses the same implementation.

# B. Experiments

## B.1. Experimental Setup Details

**Technical details** Our experimental setup utilized a system equipped with two Intel Xeon Platinum 8260 processors (48 cores total) and 384 GiB of RAM, along with 8 NVIDIA Tesla V100 GPUs, each providing 32 GiB of memory. For the calculation of the CAVs, we exclusively used the CPU-based infrastructure to ensure a consistent and comparable environment for evaluating both FastCAV and SVM-CAV. To train the ResNet50, we leveraged the full capabilities of the system.

**Models** For our experiments, we either report the metrics with respect to the last layer mentioned in the list or we average over all layers. In the case of architectures composed of repeated blocks, e.g., ResNet, we refer in the list to the final layer of each block. The notation used follows from the implementations in (Wightman, 2019) or (Wolf et al., 2019) to ensure reproducibility.

- ResNet50 (He et al., 2016): conv1, layer1, layer2, layer3, layer4

- Inception-v3 (Szegedy et al., 2016): Mixed_5b, Mixed_5c, Mixed_5d, Mixed_6a, Mixed_6b, Mixed_6c, Mixed_6d, Mixed_6e, Mixed_7a, Mixed_7b, Mixed_7c

- ConvNeXt-XXLarge (Liu et al., 2022): stem, stage1, stage2, stage3, stage4

- RegNetY (Singh et al., 2022): stem, stage1, stage2, stage3, stage4

- ViT-B/16 (Dosovitskiy et al., 2021): embeddings, encoder.block.0, encoder.block.1, encoder.block.2, encoder.block.3, encoder.block.4, encoder.block.5, encoder.block.6, encoder.block.7, encoder.block.8, encoder.block.9, encoder.block.10, encoder.block.11

- ViT-H/14-CLIP (Dosovitskiy et al., 2021; Radford et al., 2021): patch_embed, blocks.0, blocks.7, blocks.15, blocks.23, blocks.31

- EVA-G/14 (Fang et al., 2023): patch_embedding, encoder.block.0, encoder.block.9, encoder.block.19, encoder.block.29, encoder.block.39

- EVA-02-L/14 (Fang et al., 2024): patch_embedding, encoder.block.0, encoder.block.5, encoder.block.11, encoder.block.17, encoder.block.23

**Setup of $D_c$ and $D_r$** For the experiments in Section 4.1, we utilize the Broden dataset (Bau et al., 2017) to define the concept sets $D_c$. Broden describes each of its 48 pattern-based concepts with 120 images. Here, we sample 60 concept images per set $D_c$ from those 120 images once. Similarly, we sample 60 random images from ImageNet (Russakovsky et al., 2015) to define one set $D_r$ following (Kim et al., 2018).

Later, to calculate the accuracy of a computed CAV (see Equation (4)), we sample independent validation data from the remaining 60 concept examples of the Broden dataset, (Bau et al., 2017), and ImageNet (Russakovsky et al., 2015), for $D_c$ and $D_r$, respectively.

## B.2. Additional Empirical Comparisons of CAV Computation

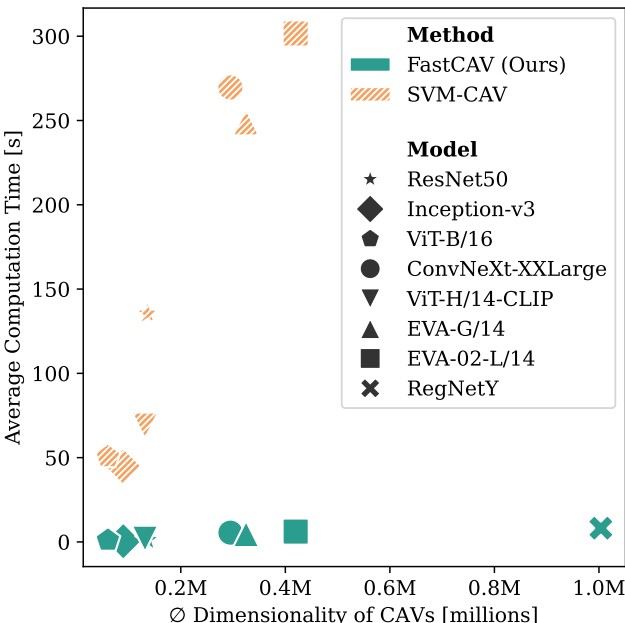

Figure 9: Comparison of computational efficiency between FastCAV and the established SVM-CAV for CAVs across different models for multiple layers. The average time taken to calculate a CAV for each method is plotted against the average dimensionality of the activation spaces, demonstrating the significant speedup achieved by our FastCAV. The differences in computation time between FastCAV and the SVM-CAV are statistically significant ($p < 0.01$) for all models. Note that SVM-CAV for the model RegNetY exceeded our maximum computational time.

As a complement to Figure 1 in our main paper, we include Figure 9, where we show the timings for all models averaged over the network layers. Note the clear difference for the RegNetY model (Singh et al., 2022) compared to Figure 1, due to the increased average activation space dimensionality. Calculating CAVs using SVMs was not feasible in our setup.

Table 4: Comparing our approach FastCAV with SVM-CAV. In this table, we focus only on the pre-final layer before the prediction, which should encode the most semantic information. **Bold values** indicate better results. "N/A" indicates that no results were produced due to the overall computational time for the respective model exceeding four days.

| | | Comp. Time [$s$] ↓ | | Accuracy ↑ | | Similarity ↑ | | |
| Model | ∅ Dim. | FastCAV | SVM-CAV | FastCAV | SVM-CAV | FastCAV | SVM-CAV | Inter-Method |
|---|---|---|---|---|---|---|---|---|
| Inception-v3 | 131,072 | **0.3**±**0.05** | 17.3±2.19 | **0.95**±**0.06** | **0.95**±**0.05** | **0.796**±**0.016** | 0.300±0.050 | 0.823±0.039 |
| ResNet50 | 100,352 | **0.3**±**0.84** | 13.8±2.04 | **0.98**±**0.04** | **0.98**±**0.04** | **0.823**±**0.010** | 0.190±0.057 | 0.775±0.045 |
| ConvNeXt-XXL | 196,608 | **1.4**±**0.10** | 45.7±4.73 | 0.96±0.05 | **0.98**±**0.04** | **0.859**±**0.031** | 0.218±0.059 | 0.816±0.042 |
| RegNetY | 1,064,448 | **6.1**±**1.04** | 312.4±63.23 | 0.97±0.05 | **0.98**±**0.03** | **0.825**±**0.009** | 0.128±0.066 | 0.675±0.065 |
| ViT-B/16 | 151,296 | **1.0**±**0.03** | 50.5±5.78 | **0.87**±**0.09** | 0.85±0.09 | **0.813**±**0.022** | 0.454±0.041 | 0.854±0.048 |
| ViT-H/14-CLIP | 328,960 | **1.8**±**0.07** | 67.0±9.67 | **0.94**±**0.06** | 0.92±0.07 | **0.654**±**0.080** | 0.328±0.084 | 0.894±0.066 |
| EVA-G/14 | 812,416 | **4.7**±**0.12** | 282.8±32.43 | **0.95**±**0.04** | 0.94±0.05 | **0.762**±**0.059** | 0.201±0.066 | 0.848±0.044 |
| EVA-02-L/14 | 1,049,600 | **6.1**±**0.14** | 252.2±29.57 | 0.93±0.07 | **0.95**±**0.05** | **0.637**±**0.075** | 0.119±0.065 | 0.747±0.059 |

Similarly, we provide Table 4 an equivalent of Table 1 containing the detailed numbers for only the pre-final layers of all networks in our study.

Both Figure 9 and Table 4 confirm the observations discussed in our main paper. However, as an additional explorative analysis, we investigate cases where FastCAV and SVM-based computation differ significantly.

### B.2.1. COMPARISON AGAINST OTHER CAV COMPUTATION APPROACHES

To provide further evidence for our claims regarding the improved computation speed, we rerun a smaller version with 30 CAVs per concept of our first experiment with additional computation methods (compare to Table 1). Specifcally, we compare against the following approaches: logistic regression (LR) (Pfau et al., 2021), sparsified logistic regression (S-LR) (Singla et al., 2021), Ridge classification as a faster alternative (Pedregosa et al., 2011), and classical LDA, which is closely related to our approach (see Section 3).

Table 5: Comparing our approach FastCAV with other CAV calculation methods regarding computation time. **Bold values** indicate better results.

| | Comp. Time [$s$] ↓ | | | | | |
| --- | --- | --- | --- | --- | --- | --- |
| Model | FastCAV (Ours) | SVM | LDA | LR | S-LR | Ridge |
| ConvNeXt-XXLarge | **0.019**± **0.013** | 1.167± 0.795 | 16.842± 11.911 | 9.360± 2.547 | 9.549± 2.734 | 0.695± 0.478 |
| EVA-02-L/14 | **0.024**± **0.001** | 1.524± 0.594 | 23.469± 3.404 | 7.659± 4.693 | 7.693± 4.731 | 0.923± 0.019 |
| EVA-G/14 | **0.018**± **0.001** | 1.109± 0.263 | 19.135± 2.200 | 8.499± 4.763 | 8.457± 4.696 | 0.681± 0.023 |
| Inception-v3 | **0.013**± **0.020** | 1.366± 0.943 | 10.755± 7.188 | 8.681± 6.076 | 8.648± 5.960 | 0.409± 0.258 |
| RegNetY | **0.039**± **0.017** | 5.207± 1.054 | 54.640± 37.392 | 11.529± 4.791 | 11.747± 4.917 | 2.184± 1.300 |
| ResNet50 | **0.016**± **0.019** | 2.908± 2.440 | 7.827± 5.601 | 8.375± 5.509 | 8.474± 5.687 | 0.314± 0.225 |
| ViT-B/16 | **0.005**± **0.002** | 0.852± 0.134 | 3.421± 0.443 | 6.517± 2.907 | 6.506± 2.909 | 0.145± 0.016 |
| ViT-H/14-CLIP | **0.008**± **0.001** | 0.456± 0.078 | 8.867± 1.037 | 7.091± 2.658 | 7.118± 2.647 | 0.288± 0.013 |

Table 6: Comparing our approach FastCAV with other CAV calculation methods regarding accuracy. **Bold values** indicate better results.

| | Accuracy ↑ | | | | | |
| --- | --- | --- | --- | --- | --- | --- |
| Model | FastCAV (Ours) | SVM | LDA | LR | S-LR | Ridge |
| ConvNeXt-XXLarge | 0.92± 0.08 | 0.95± 0.06 | 0.90± 0.15 | **0.96**± **0.05** | 0.96± 0.05 | 0.96± 0.05 |
| EVA-02-L/14 | 0.90± 0.14 | 0.90± 0.15 | 0.79± 0.20 | 0.91± 0.15 | 0.91± 0.15 | **0.91**± **0.15** |
| EVA-G/14 | 0.89± 0.15 | 0.88± 0.15 | 0.81± 0.21 | **0.90**± **0.15** | 0.90± 0.15 | 0.90± 0.15 |
| Inception-v3 | **0.95**± **0.06** | 0.93± 0.08 | 0.76± 0.19 | 0.89± 0.14 | 0.89± 0.14 | 0.89± 0.15 |
| RegNetY | 0.97± 0.05 | **0.98**± **0.03** | 0.63± 0.15 | 0.96± 0.05 | 0.96± 0.05 | 0.98± 0.04 |
| ResNet50 | 0.89± 0.15 | 0.87± 0.15 | 0.72± 0.20 | 0.92± 0.11 | 0.92± 0.11 | **0.93**± **0.12** |
| ViT-B/16 | 0.83± 0.13 | 0.81± 0.14 | **0.89**± **0.12** | 0.82± 0.14 | 0.82± 0.14 | 0.82± 0.15 |
| ViT-H/14-CLIP | 0.88± 0.15 | 0.87± 0.15 | 0.82± 0.17 | 0.89± 0.15 | **0.89**± **0.15** | 0.89± 0.16 |

Table 7: Comparing our approach FastCAV with other CAV calculation methods regarding similarity. **Bold values** indicate better results.

| | Similarity ↑ | | | | | |
| --- | --- | --- | --- | --- | --- | --- |
| Model | FastCAV (Ours) | SVM | LDA | LR | S-LR | Ridge |
| ConvNeXt-XXLarge | **0.913**± **0.014** | 0.525± 0.049 | 0.521± 0.105 | 0.672± 0.044 | 0.671± 0.044 | 0.570± 0.033 |
| EVA-02-L/14 | **0.814**± **0.020** | 0.291± 0.054 | 0.443± 0.086 | 0.685± 0.052 | 0.688± 0.050 | 0.606± 0.023 |
| EVA-G/14 | **0.789**± **0.018** | 0.331± 0.049 | 0.418± 0.070 | 0.672± 0.046 | 0.670± 0.050 | 0.608± 0.025 |
| Inception-v3 | **0.826**± **0.011** | 0.387± 0.058 | 0.132± 0.131 | 0.602± 0.037 | 0.603± 0.036 | 0.562± 0.026 |
| RegNetY | **0.825**± **0.009** | 0.128± 0.066 | 0.027± 0.115 | 0.697± 0.063 | 0.698± 0.062 | 0.675± 0.020 |
| ResNet50 | **0.791**± **0.030** | 0.398± 0.062 | 0.077± 0.159 | 0.652± 0.042 | 0.654± 0.040 | 0.602± 0.026 |
| ViT-B/16 | **0.752**± **0.029** | 0.394± 0.048 | 0.556± 0.121 | 0.592± 0.058 | 0.591± 0.059 | 0.508± 0.037 |
| ViT-H/14-CLIP | **0.749**± **0.020** | 0.342± 0.048 | 0.551± 0.149 | 0.621± 0.025 | 0.621± 0.025 | 0.595± 0.023 |

We summarize the comparison results in Tables 5 to 7. While all approaches deliver high accuracies, the logistic regression-based approaches achieve the second highest similarities after FastCAV, which coincides with increased computation times. In contrast, ridge classification offers a notable improvement in speed over SVM-based computation. However, our FastCAV still yields a substantial speed-up, while maintaining competitive performance to the more sophisticated approaches. Further, we can see that vanilla LDA is not enough to achieve similar gains. Hence, we argue that the assumptions we make in Section 3.3 are necessary under practical considerations.

B.2.2. SIGNIFICANT ACCURACY DIFFERENCES BETWEEN FASTCAV AND SVM-CAV

While we believe that our empirical evaluation provides evidence for the practicality of our approach, the strong, but necessary, assumptions (see Section 3.3) are likely not to hold in practice. Hence, we explored concrete examples where FastCAV and SVM-based computation differ in predictive performance. For example, we find in ViT-B/16 (Dosovitskiy et al., 2021), after the encoder layer 10, an accuracy difference of 40% with the SVM yielding an accuracy of 95% and FastCAV of 55%. Overall, we observed significant accuracy differences (over 25 percentage points) in 2.8% of the CAVs identified, favoring SVM in 1% and FastCAV in 1.8% of cases. In the former case, we believe that these findings are circumstances where specifically the isotropic Gaussian assumption is violated, meaning the means of $D_c$ and $D_r$ are close together in comparison to the convex hull of both sets. Overall, we find that **in 43.6% of the cases FastCAV achieves a higher accuracy, in 36.9% of the cases SVM-CAV outperforms, and in 19.5% of the cases both methods perform equally**. These observations support the findings in Section 4 in our main paper and generally favor FastCAV.

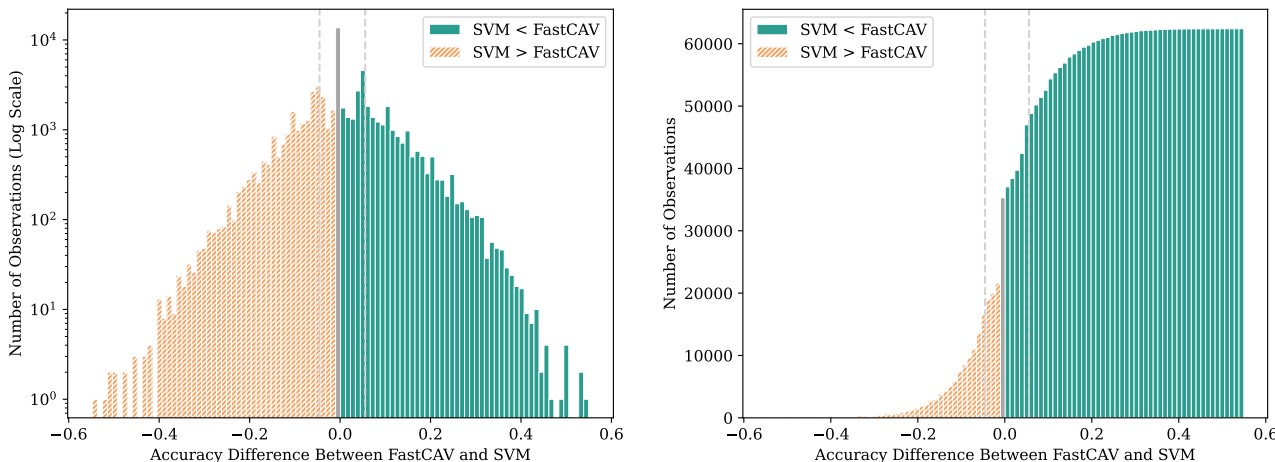

Figure 10: Left: Histogram of accuracy differences between FastCAV and SVM-CAV. Note the logarithmic y-axis scale. Right: Cumulative histogram (empirical CDF) of accuracy differences between FastCAV and SVM. Both figures show the differences for all layers of all models (see Appendix B.1). Negative values indicate that SVM-CAV achieved higher accuracy, while positive values indicate that FastCAV performed better. Agreement between both methods (a difference of zero) is colored in grey, and the grey vertical dashed lines mark the 25th and 75th percentiles (Q1 and Q3).

Next, we compare FastCAV and SVM-based computation in a more specialized task aiming to extract CAVs for medical concepts (Johnson et al., 2019; Irvin et al., 2019) to show its broad applicability.

B.2.3. CAV COMPUTATION FOR MEDICAL CONCEPTS

In our main paper, we focus on general-purpose models trained on ImageNet (Russakovsky et al., 2015) and consider a broad set of concepts (Bau et al., 2017). Here, we include an analysis of a more specialized second task. Specifically, we follow a related work exploring medical concepts (Singla et al., 2021) and train a DenseNet-121 (Huang et al., 2017) on a subset of MIMIC-CXR (Johnson et al., 2019).

Regarding training details, we use AdamW (Loshchilov & Hutter, 2017) with a learning rate of $1e-4$, a weight decay of $5e-5$, and a batch size of $64$. We train for $100$ epochs and select the model achieving the highest multi-label AUROC (0.7735) on an independent test set for our concept-based analysis.

Next, we use the CheXpert labeler (Irvin et al., 2019) to extract concept mentions from the included free-text radiology reports included with MIMIC-CXR (Johnson et al., 2019). As a set of possible concepts concerning chest x-ray images, we again follow (Singla et al., 2021).

With these extracted concepts, we compute CAVs for the four blocks of the DenseNet-121 using our FastCAV, SVM-based computation (Ghosh et al., 2023), and sparse logistic regression (Singla et al., 2021).

In our main paper (in Table 2), we report the average computation time, CAV accuracy, and intra-method similarity. Here

Table 8: Comparing our approach FastCAV with SVM-CAV and sparse logistic regression for concepts concerning chest x-ray images. **Bold values** indicate better results. Leading zeros were removed to fit the space constraints.

| Concept | Comp. Time [$s$] $\downarrow$ | | | Accuracy $\uparrow$ | | | Similarity $\uparrow$ | | |
|---|---|---|---|---|---|---|---|---|---|
| | FastCAV | S-LR | SVM | FastCAV | S-LR | SVM | FastCAV | S-LR | SVM |
| Air Bronchogram | **.007**±**.000** | 5.93±0.28 | .424±.05 | .71±.09 | **.75**±**.07** | .73±.14 | **.87**±**.02** | .49±.03 | .43±.04 |
| Ards | **.007**±**.000** | 5.97±0.54 | .391±.05 | **.82**±**.11** | .79±.07 | .76±.14 | **.89**±**.01** | .54±.04 | .48±.04 |
| Blunt | **.006**±**.001** | 6.71±0.79 | .441±.02 | **.72**±**.13** | .59±.07 | .68±.15 | **.85**±**.03** | .43±.05 | .40±.05 |
| Cardiac Silhouette | **.006**±**.000** | 5.87±0.44 | .476±.07 | .70±.07 | .69±.09 | **.71**±**.06** | **.82**±**.03** | .49±.04 | .46±.03 |
| Cephalization | **.007**±**.000** | 6.36±0.54 | .405±.04 | .72±.10 | **.80**±**.13** | .80±.06 | **.85**±**.02** | .57±.02 | .49±.03 |
| Congestion | **.006**±**.001** | 6.14±0.45 | .424±.08 | .78±.06 | .78±.07 | **.80**±**.09** | **.86**±**.02** | .48±.03 | .49±.04 |
| Heart Size | **.007**±**.000** | 6.18±0.64 | .435±.07 | .72±.10 | .73±.10 | **.76**±**.06** | **.81**±**.03** | .47±.05 | .45±.05 |
| Hilar Contour | **.007**±**.000** | 6.71±1.20 | .442±.02 | **.71**±**.10** | .69±.08 | .67±.08 | **.78**±**.02** | .42±.04 | .42±.03 |
| Hilar Engorgement | **.006**±**.001** | 6.62±0.81 | .370±.04 | .75±.10 | **.82**±**.07** | .72±.10 | **.85**±**.02** | .54±.05 | .48±.04 |
| Hilar Opacity | **.007**±**.000** | 7.01±0.62 | .451±.05 | .68±.13 | **.75**±**.06** | .63±.05 | **.79**±**.03** | .47±.05 | .43±.05 |
| Interst. Edema | **.006**±**.000** | 6.23±0.81 | .414±.07 | .82±.07 | **.83**±**.06** | .80±.05 | **.89**±**.01** | .52±.04 | .49±.04 |
| Interst. Markings | **.006**±**.001** | 6.20±0.79 | .463±.05 | .65±.09 | .70±.10 | **.71**±**.10** | **.83**±**.02** | .51±.03 | .44±.04 |
| Interst. Prominence | **.005**±**.000** | 6.72±0.64 | .427±.05 | .73±.16 | .78±.16 | **.78**±**.05** | **.86**±**.02** | .46±.03 | .42±.02 |
| Peribronch. Cuffing | **.007**±**.000** | 6.37±0.20 | .478±.07 | **.72**±**.05** | .65±.06 | .69±.13 | **.72**±**.04** | .51±.02 | .48±.02 |
| Pleural Fluid | **.006**±**.001** | 6.54±1.33 | .434±.03 | **.76**±**.13** | .72±.09 | .72±.12 | **.82**±**.02** | .45±.04 | .43±.05 |
| Vascular Marking | **.006**±**.001** | 5.74±0.87 | .439±.05 | **.81**±**.07** | .79±.09 | .74±.13 | **.85**±**.02** | .49±.04 | .46±.03 |

we provide a more detailed evaluation of the specific concepts after the dense-block 3, following (Singla et al., 2021).

Table 8 summarizes the computation time, CAV accuracies, and intra-class similarities for the set of concepts. Overall, we find that the DenseNet-121 (Huang et al., 2017) learned the specialized concepts, and we find high accuracies for all three computation methods. However, the specific accuracies per concept vary slightly. Further, we see differences in the average similarity for the three methods. FastCAV again performs best with respect to speed and similarity.

Additionally, we investigate CAV computation with respect to images containing negative results for the concepts as found by the CheXpert labeler (Irvin et al., 2019). In contrast to our expectation, we observe lower CAV accuracies in comparison to random images. However, this also holds for SVM and sparse-logistic regression-based computation, indicating a data sampling problem in the negative example regime.

Lastly, following the similar results to established methods, we wish to highlight the applicability of FastCAV for specialized downstream tasks, e.g., (Singla et al., 2021; Ghosh et al., 2023). We consider this a valuable direction for future work.

### B.3. TCAV — Additional Results

In our main paper, we include the results for TCAV (Kim et al., 2018) for the class *ladybug* using both our FastCAV as well as SVM-based CAV computation. To further strengthen our analysis, we follow (Kim et al., 2018) and evaluate additional qualitative examples. Here, we follow the setup detailed in Section 4.3 and use a GoogleNet (Szegedy et al., 2015) trained on ImageNet (Russakovsky et al., 2015). We visualize the TCAV scores for the same layers selected in (Kim et al., 2018).

Figure 11, Figure 12, Figure 13, and Figure 14 display classes *ladybug*, *fire-engine*, *police-van*, and *zebra*, respectively. In all cases, we find only small differences between our CAV computation and the established SVM-based approach. Similar to our discussion in the main paper, we find lower variances using FastCAV. Nevertheless, the overarching trends are consistent between both methods. Further, the resulting insights into the GoogleNet model align well with (Kim et al., 2018). For example, for both methods, we find the concept "red" to be most important for the classes *ladybug* and *fire-engine*. Similarly, we observe "blue" influencing class *police-van* and "striped" influencing class *zebra*. This again demonstrates that both FastCAV and SVM-CAV can be used to generate valid concept-based explanations, supporting the claims made in our main paper.

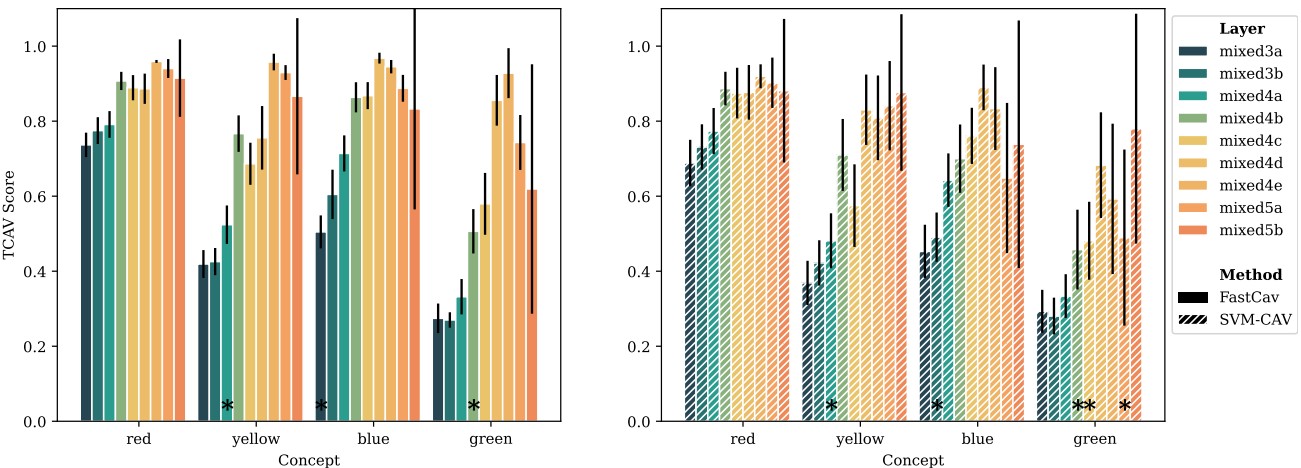

Figure 11: Example of the concepts "red", "yellow", "blue" and "green" for class *ladybug* using TCAV. We follow (Kim et al., 2018) and mark CAVs that are not statistically significant with "*".

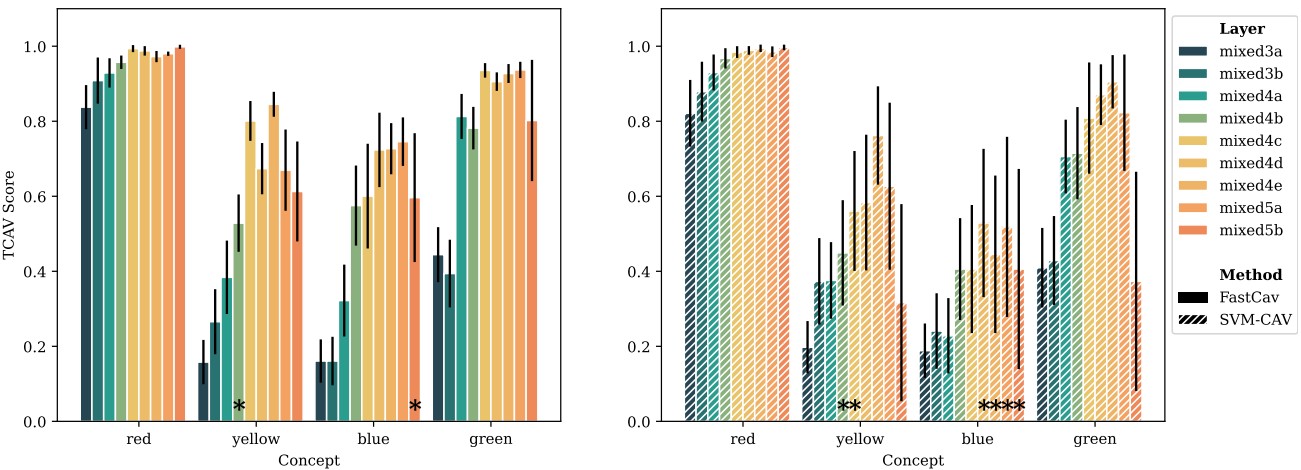

Figure 12: Example of the concepts "red", "yellow", "blue" and "green" for class *fire-engine* using TCAV. We follow (Kim et al., 2018) and mark CAVs that are not statistically significant with "*".

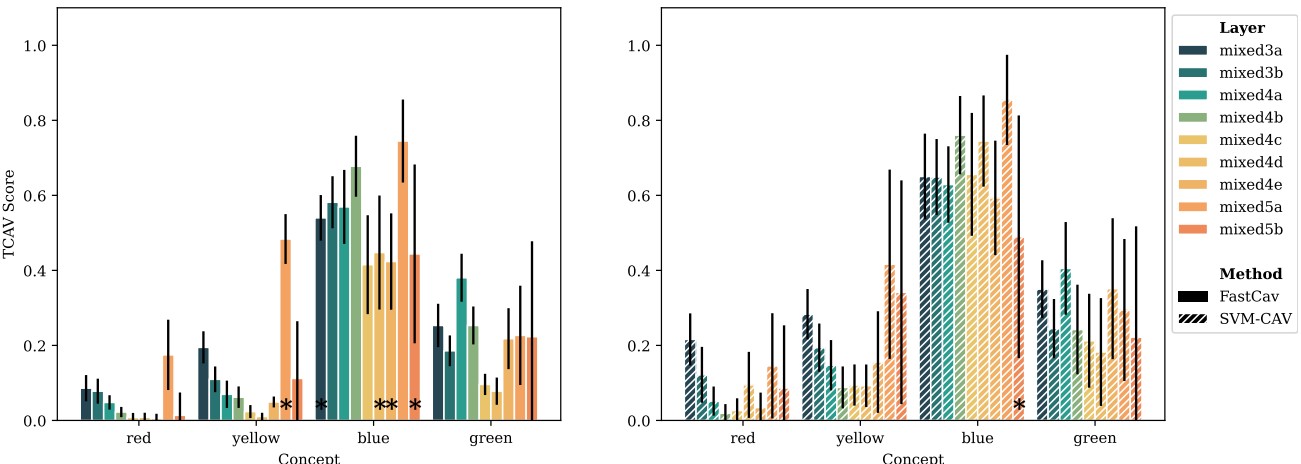

Figure 13: Example of the concepts "red", "yellow", "blue" and "green" for class *police-van* using TCAV. We follow (Kim et al., 2018) and mark CAVs that are not statistically significant with "*".

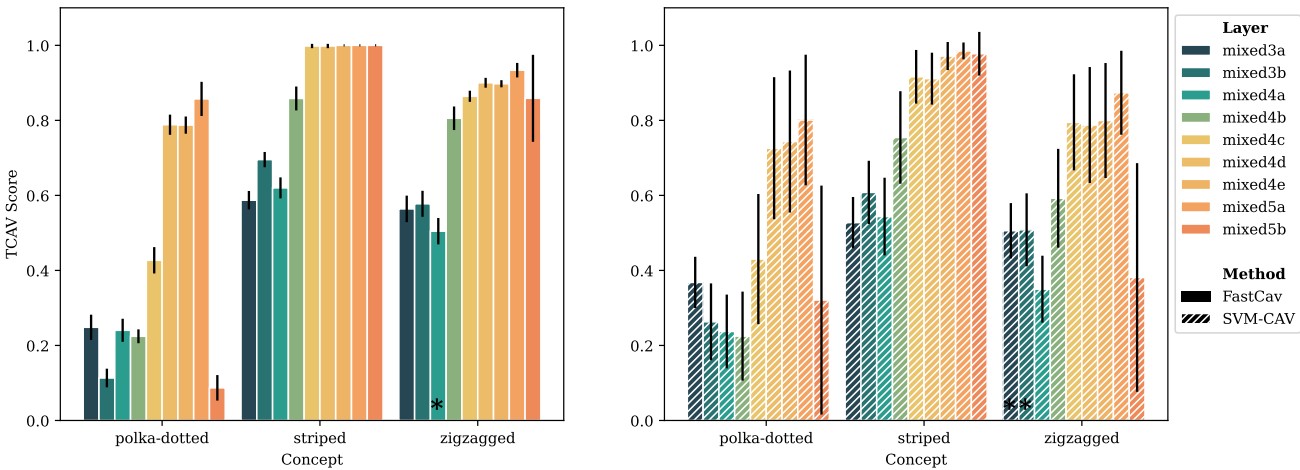

Figure 14: Example of the concepts "polka-dotted", "striped" and "zigzagged" for class *zebra* using TCAV. We follow (Kim et al., 2018) and mark CAVs that are not statistically significant with "*".

Table 9: The TCAV scores (Kim et al., 2018) for the four most salient concepts discovered in an ImageNet (Russakovsky et al., 2015) trained Inception-v3 (Szegedy et al., 2016) model using ACE (Ghorbani et al., 2019). We list scores for the classes *Lionfish*, *Zebra*, and *Police Van*.

| Concept | *Lionfish* | | *Zebra* | | *Police Van* | |
|---|---|---|---|---|---|---|
| | FastCAV | SVM-CAV | FastCAV | SVM-CAV | FastCAV | SVM-CAV |
| 1 | 0.78 | 0.73 | 0.68 | 0.66 | 0.88 | 0.78 |
| 2 | 0.75 | 0.73 | 0.67 | 0.61 | 0.86 | 0.77 |
| 3 | 0.69 | 0.67 | 0.60 | 0.58 | 0.83 | 0.76 |
| 4 | 0.68 | 0.67 | 0.59 | 0.56 | 0.80 | 0.70 |

## B.4. ACE — Additional Results

In our main paper, we visualize and compare the most salient concepts for the class *Lionfish* found by ACE with FastCAV and SVM-CAV. Here, we add to this and visualize the other classes selected in (Ghorbani et al., 2019). For class *Lionfish*, ACE using FastCAV finds 22 relevant concepts, and with SVM-CAV, we identify 24 overall. For *Zebra*, we find 19 concepts using our approach and 18 with SVM computation. Lastly, for class *Police Van*, we identify 24 and 22, respectively. In Table 9, we list the TCAV scores of the four most salient concepts for each class, respectively.

In Figure 15, Figure 16, and Figure 17, we display the three most salient concepts for both methodologies. In all cases, we observe similar results for both FastCAV and SVM-CAV, supporting our findings in the main paper.

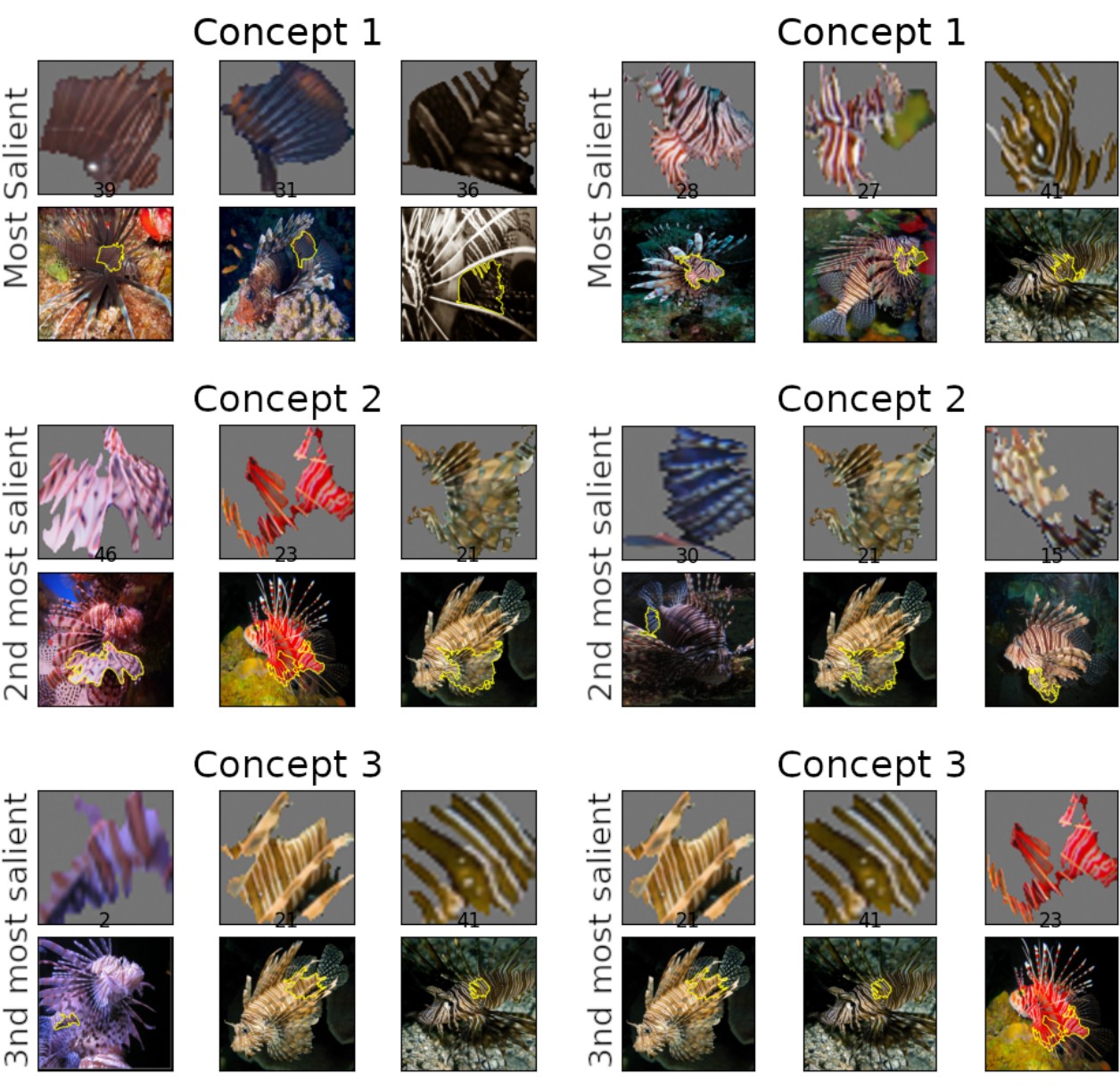

(a) FastCAV based ACE (Ghorbani et al., 2019).    (b) SVM-CAV based ACE (Ghorbani et al., 2019).

Figure 15: Comparison of the most salient concepts discovered by ACE (Ghorbani et al., 2019) using either our FastCAV or the established SVM-CAV. Here, we use class *lionfish* and display the three most salient concepts. We find the discovered patches between both approaches similar and congruent with the original observation in (Ghorbani et al., 2019).

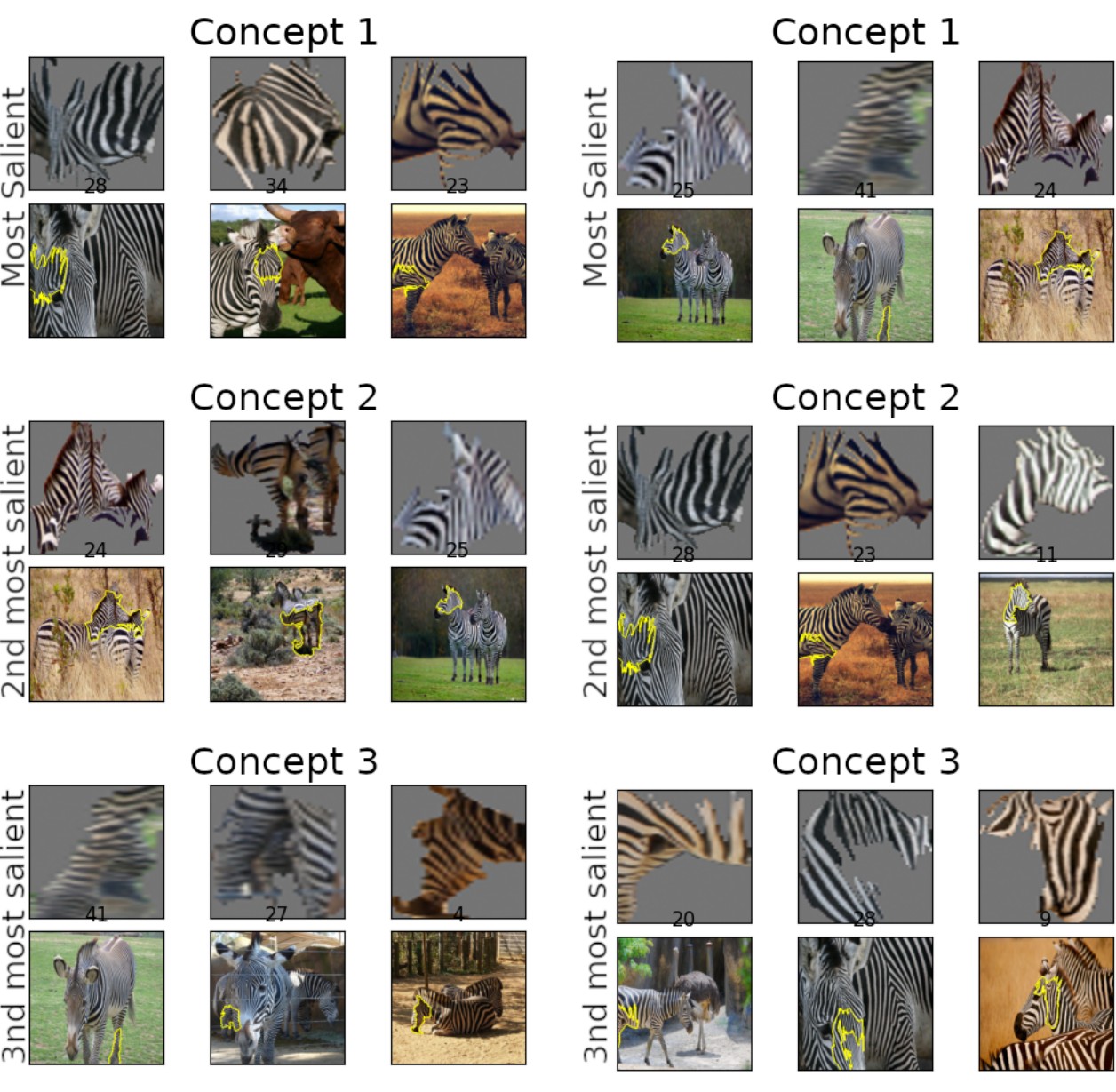

(a) FastCAV based ACE (Ghorbani et al., 2019).  (b) SVM-CAV based ACE (Ghorbani et al., 2019).

Figure 16: Comparison of the most salient concepts discovered by ACE (Ghorbani et al., 2019) using either our FastCAV or the established SVM-CAV. Here, we use class *zebra* and display the three most salient concepts. We find the discovered patches between both approaches similar and congruent with the original observation in (Ghorbani et al., 2019).

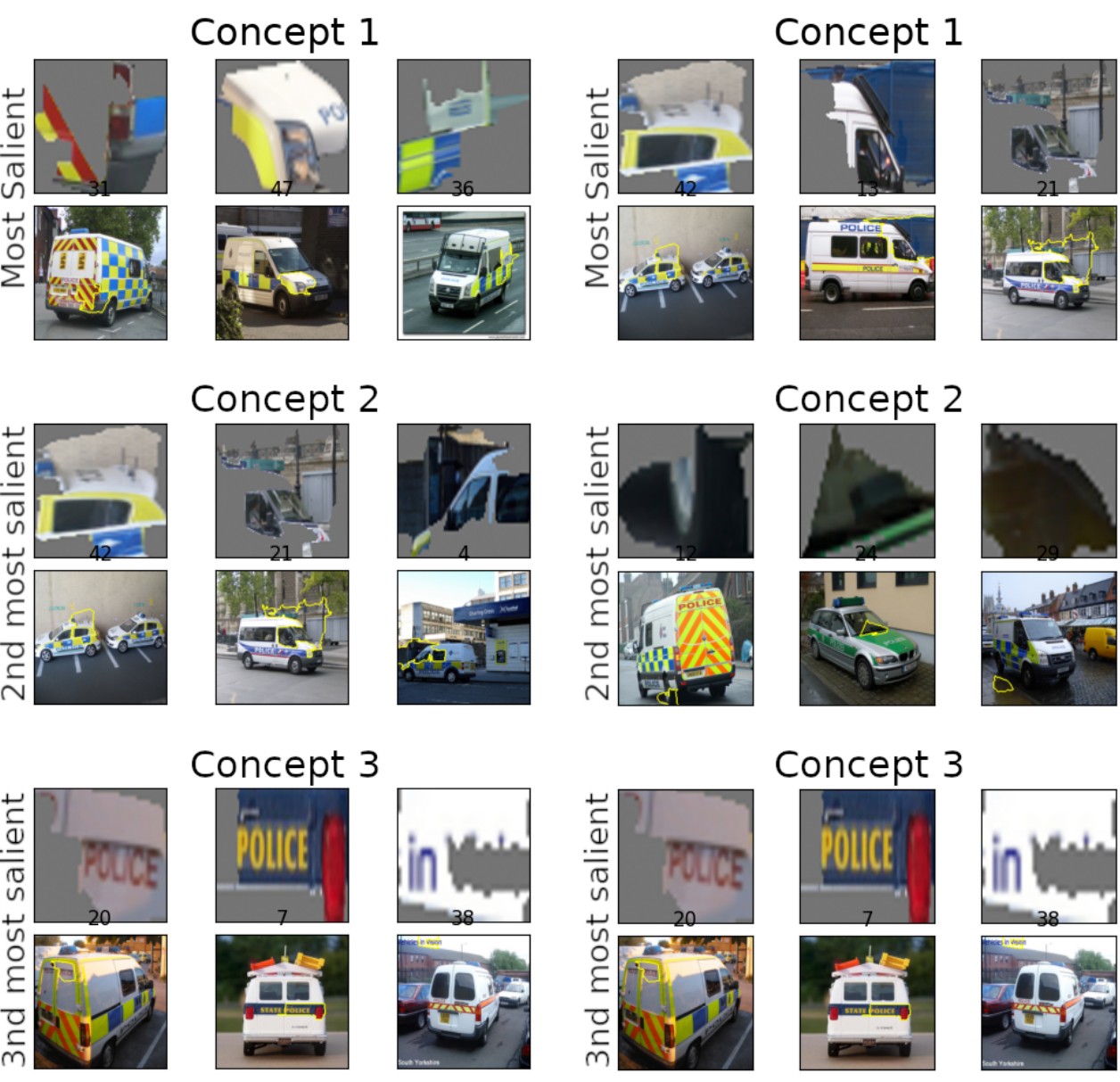

(a) FastCAV based ACE (Ghorbani et al., 2019).

(b) SVM-CAV based ACE (Ghorbani et al., 2019).

Figure 17: Comparison of the most salient concepts discovered by ACE (Ghorbani et al., 2019) using either our FastCAV or the established SVM-CAV. Here, we use class *police* and display the three most salient concepts. We find the discovered patches between both approaches similar and congruent with the original observation in (Ghorbani et al., 2019).

### B.5. Tracking CAVs During Training — Additional Visualizations

**Setup Details:** We train a ResNet50 (He et al., 2016) model on the ImageNet 2012 dataset (Russakovsky et al., 2015) using a batch size of 256 for 90 epochs following (Paszke et al., 2019). Each image is resized to 256 pixels, followed by a random crop of 224 by 224, a random horizontal flip, and normalization with mean values of 0.485, 0.456, and 0.406 and standard deviations of 0.229, 0.224, and 0.225. The optimizer is Stochastic Gradient Descent with an initial learning rate of 0.1, momentum of 0.9, weight decay of 1e-4, and a StepLR scheduler with a step size of 30 epochs and a gamma of 0.1.

**Additional Visualizations:** In our main paper (Figure 6), we visualize the average CAV accuracies during the training of a ResNet50 (He et al., 2016). We calculate this average over the concepts included in (Bau et al., 2017). In Figure 18, we visualize the same averages but additionally include lines for all individual concepts to highlight the variances.

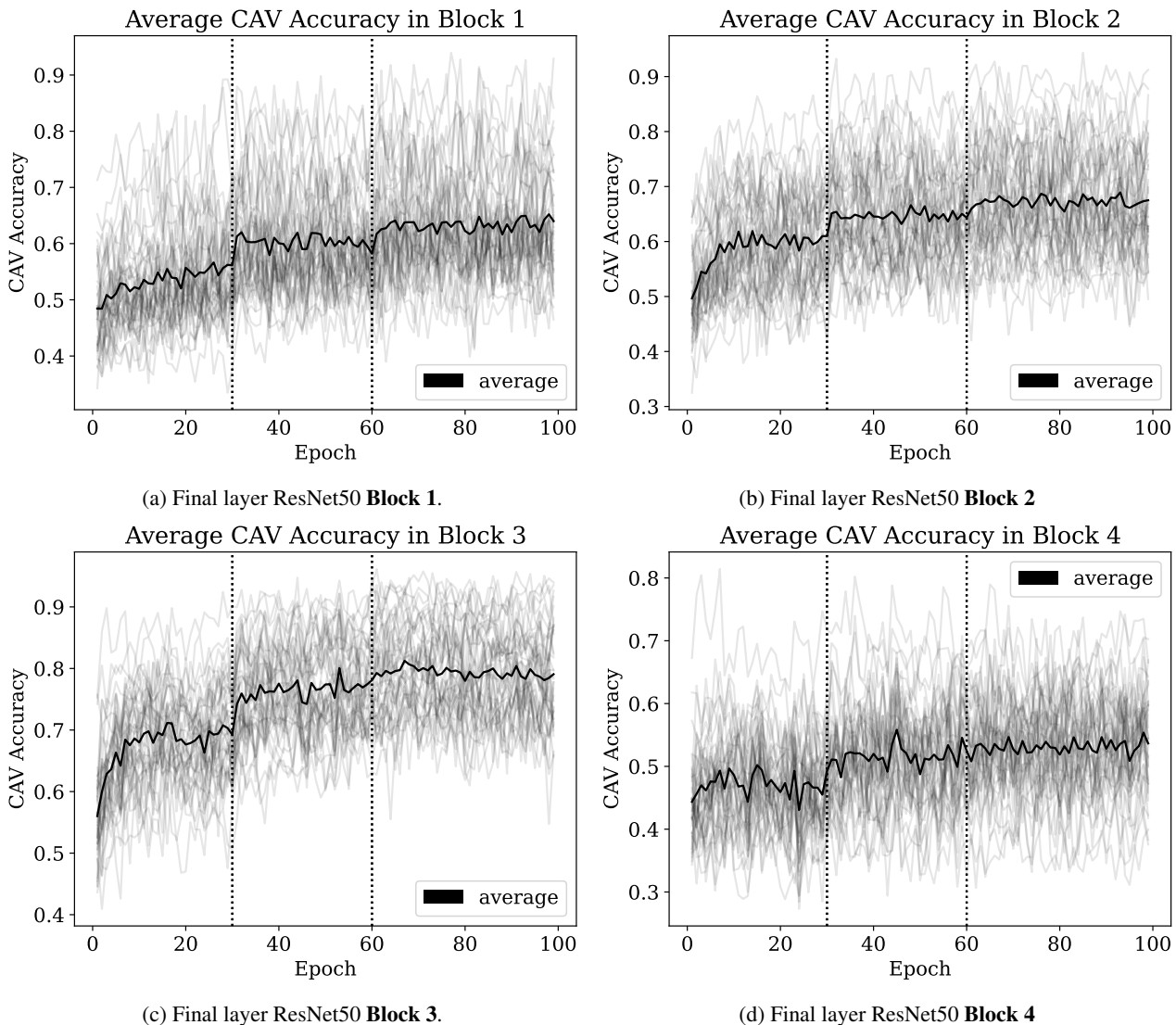

(a) Final layer ResNet50 **Block 1**.

(b) Final layer ResNet50 **Block 2**

(c) Final layer ResNet50 **Block 3**.

(d) Final layer ResNet50 **Block 4**

Figure 18: Additional visualizations pertaining to the lefthand side in Figure 6. Specifically, we visualize the average accuracies achieved by CAVs after the final layers in each of the four ResNet blocks (Figure 18a - Figure 18d) for the concepts in (Bau et al., 2017). Here, we display the changes for all concepts to showcase the respective variances during training. Vertical dotted lines again correspond to epochs where the learning rate was divided by ten.

Similarly, following our qualitative selection of specific concepts for the final layer of block 3 in Figure 6 (middle plot), we include visualizations for the other blocks in Figure 19. We confirm that the observations made in the main paper similarly

hold for the other layers. In addition, we find the lowest CAV accuracies in Figure 19d, which is consistent with the results for the development of the average concept.

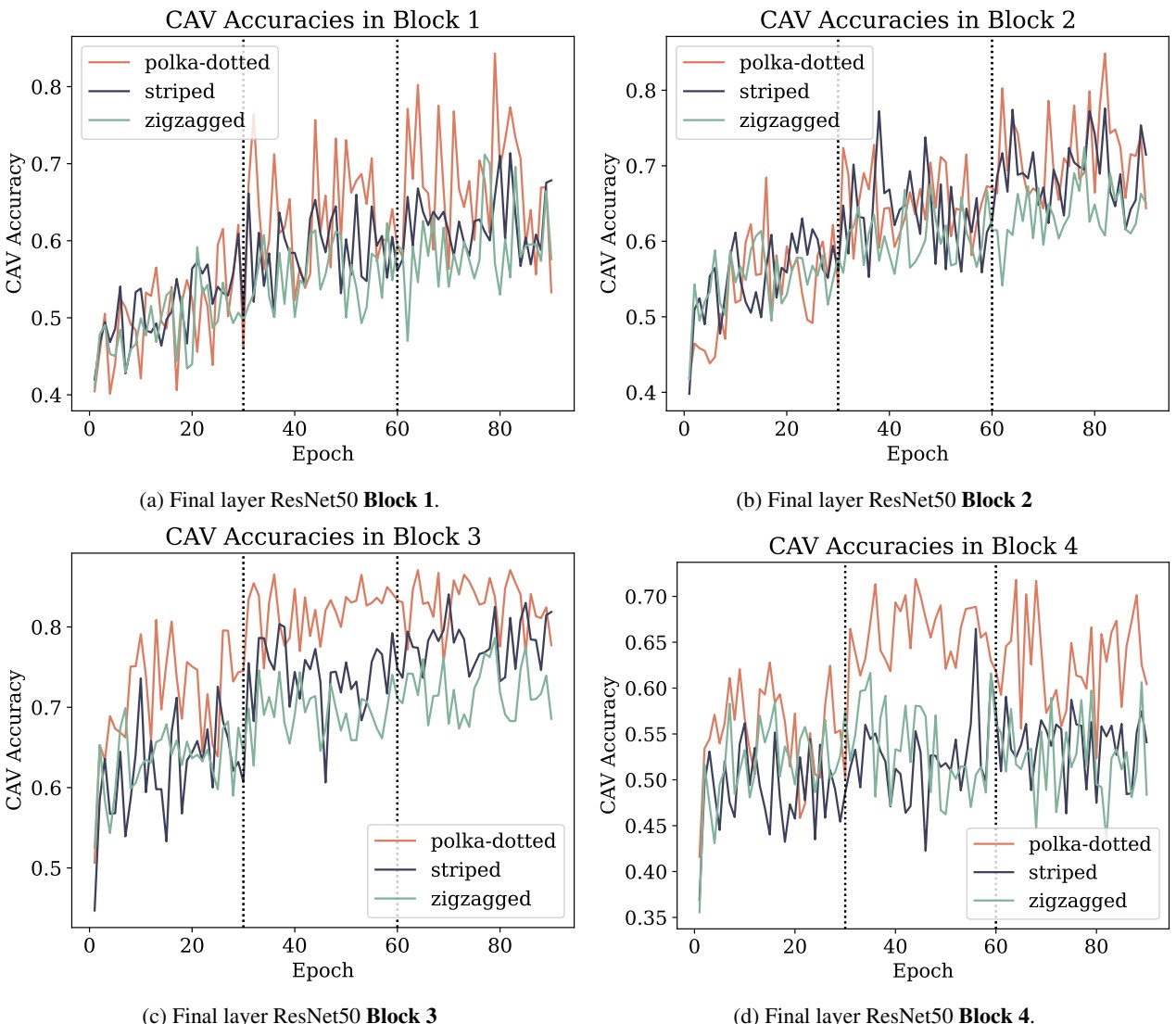

(a) Final layer ResNet50 **Block 1**.

(b) Final layer ResNet50 **Block 2**

(c) Final layer ResNet50 **Block 3**

(d) Final layer ResNet50 **Block 4**.

Figure 19: Additional visualizations for the middle plot in Figure 6. Specifically, we qualitatively visualize the accuracies of selected CAVs after the final layers in each of the four ResNet blocks (Figure 19a - Figure 19d) for the concepts in (Bau et al., 2017). Figure 19c is equivalent to the selected version in Figure 6. Vertical dotted lines again correspond to epochs where the learning rate was divided by ten.

**Additional Explorative Analysis**    To expand our discussion regarding the training analysis (see Section 4.5 and above), we provide a more comprehensive exploration of the potential of FastCAV.

First, in our main paper, we include results ranking concepts by their area under the curve (AUC) during training and across layers. This reveals that the learnability/difficulty of concepts can vary wildly across layers, with some concepts that are quickly learned in early layers becoming more difficult to learn in later layers.

Here, we go one step further and evaluate the ratio of learned concepts per layer during the training. In Figure 20, we summarize the corresponding results. We find that during the training, the ratio of concepts learned increases, indicating that the model learns task-relevant concepts. These findings are congruent with previous analysis of neural network training dynamics, e.g., (Shwartz-Ziv & Tishby, 2017; Penzel et al., 2022). Further, we observe the highest ratios during the complete

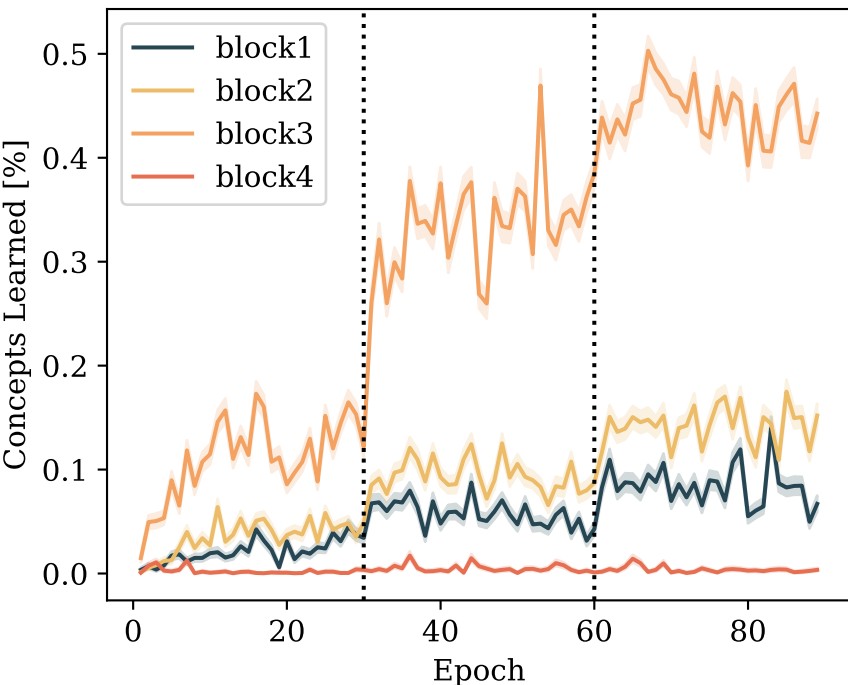

Figure 20: Percentage of learned concepts by CAVs and block during training of a ResNet50 (He et al., 2016) on ImageNet (Russakovsky et al., 2015).

training process for the architecture block 3. Given that our probing dataset focuses on the textures, this observation aligns with findings made in (Bau et al., 2017).

Overall, we hope that by providing a faster and more efficient concept extraction procedure with FastCAV, we can empower researchers to explore more complex scenarios and larger concept sets, potentially uncovering novel phenomena that were previously inaccessible due to computational constraints.

