# OpenReview forum: "FastCAV: Efficient Computation of Concept Activation Vectors for Explaining Deep Neural Networks"
_ICML.cc/2025/Conference — ICML 2025 poster_

### Official Review · Reviewer_KdBw · 2025-03-13

**Overall Recommendation:** 3

**Summary:**

The paper describes the method for accelerating the computation of concept activation vectors, which uses, instead of the SVM linear classifier, the Fisher's linear discriminant analysis (LDA).

## After the rebuttal

Many thanks to the authors for the work and a good rebuttal discussion. The authors address my main concerns, therefore I am increasing my score.

The main question was about the contribution of this work in connection to the Fisher's LDA. The authors clarified upon this work's motivation and the outline of the contribution, and would be expected to revise the motivation accordingly, including describing it in the beginning of Section 3.

The answer to the question about new insights is also reasonable.

I have checked the responses to the other rebuttals, and I believe the authors addressed the points accordingly as well.

New experiments as per request from Reviewer ULp4 would also be a good addition to the paper.

**Claims And Evidence:**

The analysis appears to be correct, and the method and analysis appears to be sound, however it is important to see it in the wider context.

The authors claim:
"We introduce FastCAV, a novel approach for computing Concept Activation Vectors (CAVs) that significantly reduces computation time by up to 63.6×, making it more efficient and scalable."

The main concern is that the authors need to justify the novelty of the approach. As the description states now, it does not seem to be a combination of CAVs with LDA, both well-known.

"• We provide a theoretical foundation for FastCAV and specify concrete assumptions under which it is equivalent to other linear methods."
The problem with the claim might be that I understand that the theoretical argument coincides with the LDA description as per (Bishop, 2006) and ultimately going back to Fisher (1936) as cited in the paper.

Fisher, R. A. (1936). "The Use of Multiple Measurements in Taxonomic Problems"

•" We demonstrate empirically that FastCAV produces similar CAVs to the established SVM-based approach, leading to comparable insights in downstream methods, e.g.,(Kim et al., 2018; Ghorbani et al., 2019)."
This claim appears to be matched with the evidence, as described below in the review of experimental design and analyses.

•" Using FastCAV, we investigate the evolution of concepts during training and across different layers of a neural network."
It is true that the evolution of concepts is investigated in Section 4.4, however it is not by any measure sufficient as the authors just present the plot of CAV accuracy depending on accuracy.

**Essential References Not Discussed:**

I believe the references are sufficiently discussed.

**Experimental Designs Or Analyses:**

The experiment underpin the claims.
The authors compare the methods against the standard CAV, as well as other baselines on computational time as well as performance. They find that they obtain competitive performance while substantial (on average 46.4×) acceleration compared to the baseline methods.

Table 1 does not contain the confidence intervals, could the authors add them.

**Methods And Evaluation Criteria:**

The proposed method seems to be a justified solution for the given problem, which is accelerating CAV methods, however, the authors need to emphasise in which way it is not a straightforward combination of Fisher (1936) and the original CAV.

I don't think the ablation study, proposed in Section 4.3, is an ablation study, which involves removal or damage to the components (see e.g., Meyes et al (2019). The hyper-parameter value influence studies would be sensitivity analysis.

Meyes et al (2019) Ablation Studies in Artificial Neural Networks

Section 4.4 proposes only a rudimentary analysis of leveraging this low-resource method for interpreting the training process online, which involves the CAV accuracy tracking over time. In my view, this part actually highlights significance of the contribution the most and could be analysed in terms of how that can help provide diagnosis tools for neural network training this can involve quantitative and qualitative evaluation.

**Other Comments Or Suggestions:**

No other suggestions

**Other Strengths And Weaknesses:**

Strengths:
Correctness: the analysis appears to be correct
Soundness: the method and analysis appears to be sound

Weaknesses:
-originality: it is important that the authors clarify upon the contribution. Right now, it seems to be that the gist of the proposed approach constitutes replacing linear SVM with the Fisher’s LDA and the evaluation aiming to show its competitive accuracy and computational time. However, my main concern is that this core of a contribution might not give enough new insight into the CAVs as it only seems intuitive that SVM can be replaced with a linear classifier. I suspect though that the authors could describe the motivation more clearly, perhaps putting emphasis on the new ways of explanations that it could help achieve which the state-of-the-art cannot (e.g., explaining the training dynamics which isn’t computationally feasible otherwise). At this stage, the paper merely hints at it at Section 4.4, and I would think it could be strenghened.

**Questions For Authors:**

The main sticking point is the contribution. The contribution seem to be a drop-in replacement of SVM with Fisher LDA, which may fall short of the expectations of the venue. Therefore, it is crucial that the authors confirm in the rebuttal and revision why this contribution is novel and not just a combination of two well-known methods.

The other problem is the theoretical analysis, which seems to be centred around the properties of the Gaussian distribution as per Bishop (2006), underpinning LDA (see Sections 3.2, 3.3, Appendix A.1).

**Relation To Broader Scientific Literature:**

The main works, in my opinion, relate to Fisher LDA and concept activation vectors. Both aspects have been covered in the related works section.

**Theoretical Claims:**

I have checked the theoretical claims, and I believe they are entirely mirroring Bishop (2006), which is duly cited. The authors need to emphasise in which way this would be a novel contribution.

---

> ### Author Rebuttal · Authors · 2025-03-31
>
> We thank the reviewer for their effort in checking our theoretical contributions and providing insightful comments. Particularly, we appreciate the reviewer for highlighting the correctness and soundness of our analysis.
>
> ## Implemented Changes
> We agree with and follow the suggestion of the reviewer and include standard deviations for the accuracies in our Table 1 (compare to reviewer ULp4).
> Additionally, we add an experiment on chest radiographs to further validate our approach (see reviewer ADni) and showcase an additional use-case.
> Finally, we appreciate the correction regarding our sensitivity analysis and will improve the terminology in Section 4.3 to align with Meyes et al. (2019).
>
> ## Contribution and Comparison to Fisher’s LDA
> The reviewers main concern is our contribution in connection to an LDA. We welcome the opportunity to clarify these points. Overall, we see our contribution in establishing a more efficient alternative to other CAV calculations that enables researchers to solve more complex and demanding tasks due to the speed-up of the calculation itself.
> The specific point of the reviewer is that our contribution is to “[replace] linear SVM with the Fisher’s LDA”.  We want to emphasize that we derive FastCAV based on the orthogonality of concepts in feature space observed previously in (Olah et al., 2020, Elhage et al., 2022) for the application of identifying feature directions in a model’s activation space. We then show in Section 3.3 that the resulting calculation is equivalent in expectation to the solution of a Fisher’s LDA only under isotropic, equally mixed, Gaussian assumptions [L179 right-hand side].
> To further emphasize the practical importance of these assumptions, we add an empirical comparison between FastCAV and LDA. We observe significantly lower computation time for FastCAV (see table for reviewer ULp4). Further, Fisher’s LDA is on average slower than the established SVM-based calculation. Thus, it is non-trivial to select the correct classifier and assumptions for accelerated CAV calculation.
>
> The theoretical equivalence to Fisher’s LDA under the described assumptions is necessary to connect our approach to the predominant SVM-based calculation. Particularly, while previous works discuss the connection of Fisher’s LDA to the decision boundary found by an SVM (Shashua 1999), we additionally provide evidence (Appendix A.2) that the required circumstances apply to CAV computation in a model’s activation space. Specifically we show empirically that for CAV calculation there is a high ratio of support vectors among $D_r \cup D_c$. Hence, the connection and application to concept discovery and comparison to existing SVM-based computation is a novel contribution.
>
> Following the feedback of the reviewer, we will improve our motivation in the beginning of Section 3.
>
> ## New Insights Unlocked by FastCAV
> We appreciate the reviewer for highlighting FastCAV's acceleration and its application potential as a key contribution. Acting on their suggestion, we expand our discussion regarding the training analysis (Experiment 4) to provide a more comprehensive exploration of the potential.
>
> 1. We conduct an analysis ranking concepts by their area under the curve (AUC) during training and across layers. This approach reveals that the learnability/difficulty of concepts can vary wildly across layers, with some concepts that are quickly learned in early layers becoming more difficult to learn in later layers. For instance, the concept 'wave' is the most readily learned concept in the first block, but its ranking deteriorates to second after block two and ultimately drops to 32nd in the final block. In contrast, the concept 'lined' remains consistently high ranked across all layers. We will include visualizations and additional discussion in our revised paper.
>
> 2. We evaluate the ratio of concepts learned in each layer over the training process. We find that during the training this ratio increases indicating that the model learns task-relevant concepts. This result is congruent with previous analysis of neural network training dynamics, e.g., Shwartz-Ziv and Tsihby, 2017, “Opening the Black Box of Deep Neural Networks via Information”. However, the speed-up of Fast-CAV enables an analysis of the training dynamics on the level of human-interpretable concepts.
>
> Overall, by providing a faster and more efficient procedure, we expect to empower researchers to explore more complex scenarios and larger concept sets, potentially uncovering novel phenomena that were previously inaccessible due to computational constraints.
> Finally, we are eager to address any further questions and discuss additional points.

---

> > ### Comment · Reviewer_KdBw · 2025-04-07
> >
> > The authors address my concerns with the rebuttal, therefore I am increasing my score.
> >
> > The main question was about the contribution of this work in connection to the Fisher's LDA.  The authors clarify upon the motivation of this work and the outline of the contribution, and would be expected to revise the motivation accordingly, including describing it in the beginning of Section 3.
> >
> > The answer to the question about new insights is also reasonable.
> >
> > I have checked the responses to the other rebuttals, and I believe the authors addressed the points accordingly as well.
> >
> > New experiments as per request from Reviewer ULp4 would also be a good addition to the paper.

---

> > > ### Author Response · Authors · 2025-04-07
> > >
> > > We thank the reviewer for their encouraging feedback. Following the suggestions, we will ensure to revise our motivation accordingly. We also appreciate the reviewer's consideration of our other responses.

---

### Official Review · Reviewer_ADni · 2025-03-14

**Overall Recommendation:** 4

**Summary:**

The paper introduces FastCAV, a method to compute Concept Activation Vectors (CAVs) up to 63.6× faster than traditional SVM-based approaches by leveraging simple mean vector computations under theoretical assumptions. FastCAV achieves comparable accuracy and interpretability to existing methods while significantly reducing computation time, making it practical for analyzing large modern neural networks. The authors validate FastCAV with extensive experiments on multiple architectures and demonstrate its use in tracking concept evolution during model training.

**Claims And Evidence:**

The paper’s main claims are that (1) FastCAV is up to 63.6× faster than SVM-based CAV computation while maintaining comparable quality, and (2) FastCAV is theoretically justifiable under assumptions of isotropic Gaussian distributions for concept and random images. Overall, these claims are supported by experimental results and references to linear discriminant analysis. However, there are areas where the evidence could be improved:
1. While the authors do show that FastCAV recovers near-identical directions as an SVM in high-dimensional settings, it would have been helpful to see more thorough statistical significance tests, or a direct demonstration of how violations of the Gaussian or isotropy assumptions degrade performance.
2. The experiments rely heavily on ImageNet-based networks and a set of curated concept images. The results would be more convincing if tested on diverse tasks (e.g., medical or multimodal applications). For medical imaging, the authors can look into this paper for concept extraction:
[1] Using Causal Analysis for Conceptual Deep Learning Explanation. Singla et al. MICCAI 2021
[2] Distilling BlackBox to Interpretable models for Efficient Transfer Learning. Ghosh et al. MICCAI 2023.

**Essential References Not Discussed:**

NA

**Experimental Designs Or Analyses:**

The experiments are fairly comprehensive.

**Methods And Evaluation Criteria:**

The proposed method (FastCAV) is coherent in the sense that it directly addresses the high computational overhead of linear SVM training in high dimensions. One limitation is that the paper focuses mostly on speedups and accuracy for separating concept examples from random images. It might have been valuable to include interpretability-specific user studies (i.e., whether actual users found FastCAV-based explanations equally comprehensible or trustworthy).

**Other Comments Or Suggestions:**

1. Can the authors investigate cases where SVM-based CAVs and FastCAV disagree on concept classification?
2. Showing how FastCAV-based results compare if, instead of random images, “negative concept” images are used. That could confirm whether orthogonality assumptions hold better in different negative sampling regimes.

**Other Strengths And Weaknesses:**

1. The method’s stability and accuracy under distribution shifts or adversarial perturbations remain unclear.
2. The notion that linear directions suffice to encode complex concepts (particularly in Transformers) is an assumption that might need further empirical justification.

**Questions For Authors:**

Have you seen any cases where FastCAV completely fails (e.g., for extremely rare or ill-defined concepts)? How might such failure modes be detected automatically?

**Relation To Broader Scientific Literature:**

The paper is well-situated in prior work on concept-based explanations (particularly CAV/TCAV) and the notion of linear separability in neural activations.

**Theoretical Claims:**

The authors reference Fisher discriminant analysis and prior results connecting Fisher’s linear discriminant to linear SVMs. A weaker point is that the equivalence rests on fairly strong assumptions (isotropy and equal class priors). While this is not uncommon in theoretical exposition, the paper could have discussed more thoroughly how real-world deviations from isotropy (e.g., extremely skewed concept distributions, unusual image statistics) might reduce the fidelity of FastCAV.

---

> ### Author Rebuttal · Authors · 2025-03-31
>
> We thank the reviewer for their positive feedback and appreciate that they see our contribution well-situated within prior work. We are glad that our empirical comparison to SVM-based CAV computation was found to be fairly comprehensive. Additionally, we agree with the reviewer that more applications further validate the effectiveness of FastCAV.
>
> ## Additional Medical Application
> The reviewer provided references concerning a medical task classifying chest radiographs. We take up this proposal and follow (Singla et al. 2021) to train a DenseNet-121 on a smaller subset of MIMIC-CXR due to time constraints. Next, we use the CheXpert labeler to extract concept mentions from the included free-text radiology reports. Specifically, we take the concepts specified in Figure 3 in (Singla et al. 2021).
>
> Using these concepts, we compute CAVs for the four blocks of the DenseNet-121 using our FastCAV, SVM-based computation (Gosh et al. 2023), and sparse logistic regression (Singla et al. 2021). Here, we include a comparison of the average computation time and the achieved CAV accuracies, which we will also add in our appendix in the revised version:
>
> |Model|Method|Comp. Time [s]↓|Accuracy↑|Similarity↑|
> |---|---|---|---|---|
> |DenseNet-121|FastCAV (Ours)|**0.006±0.001**|0.715±0.111|**0.793±0.030**|
> |DenseNet-121|Sparse-Logistic Regression|6.370±0.756|**0.721±0.129**|0.500±0.036|
> |DenseNet-121|SVM|0.439±0.071|0.709±0.125|0.457±0.038|
>
> We observe performance similar to that of the established approaches in this domain. However, note the strongly reduced computation time for FastCAV. Additionally, we report the achieved accuracies for selected concepts after dense-block 3 following (Singla et al. 2021) and will add the full results in our updated revision:
>
> |Method|Concept|Comp. Time [s]↓|Accuracy↑|Similarity↑|
> |---|---|---|---|---|
> |FastCAV (Ours)|Cardiac Silhouette|**0.006±0.000**|0.695±0.068|**0.820±0.031**|
> |Sparse-Logistic Regression|Cardiac Silhouette|5.866±0.437|0.688±0.086|0.491±0.041|
> |SVM|Cardiac Silhouette|0.476±0.068|**0.710±0.057**|0.459±0.026|
> ||||||
> |FastCAV (Ours)|Congestion|**0.006±0.001**|0.781±0.062|**0.855±0.020**|
> |Sparse-Logistic Regression|Congestion|6.137±0.448|0.778±0.073|0.484±0.032|
> |SVM|Congestion|0.424±0.079|**0.796±0.091**|0.490±0.043|
> ||||||
> |FastCAV (Ours)|Interstitial Edema|**0.006±0.000**|0.817±0.074|**0.886±0.014**|
> |Sparse-Logistic Regression|Interstitial Edema|6.230±0.806|**0.831±0.058**|0.516±0.043|
> |SVM|Interstitial Edema|0.414±0.072|0.798±0.053|0.490±0.043|
>
>
> We can see that the DenseNet-121 learned the shown concepts. However, the achieved CAV accuracies vary slightly. Further, we see differences in the average similarity for the three methods. FastCAV performs best with respect to speed and similarity.
>
> Additionally, we follow the suggestion of the reviewer and compute CAVs with respect to images containing negative results for the concepts as found by the CheXpert labeler. In contrast to our expectation, we observe lower CAV accuracies in comparison to random images. However, this holds also for SVM and sparse-logistic regression based computation indicating a data sampling problem in the negative example regime.
>
> Finally, while we recognize the importance and are interested in evaluating the related downstream tasks (Singla et al. 2021, Gosh et al. 2023), the time constraints of the rebuttal period limited the scope of our investigation. We consider this a valuable direction for future work.
>
>
> ## Differences in FastCAV and SVM-based Computation
> We appreciate the reviewer remarks on the theoretical connection of our approach to the existing SVM approach, going over Fisher’s LDA and strong assumptions [L175 right-hand side]. While we agree with the reviewer that these strong assumptions are likely to be violated in practice, we believe that our empirical evaluation in Section 4.1 and the medical application above provide evidence for the practicality of our approach.
>
> Nevertheless, following the reviewer's comments, we investigated examples where FastCAV and SVM-based computation differ. For example, in ViT-B/16 after the encoder layer 10 we observe an accuracy difference of 40% with the SVM yielding an accuracy of 95% and FastCAV of 55%. Overall, we observed significant accuracy differences (over 25%) in 2.8% of the CAVs identified, favoring SVM in 1% and FastCAV in 1.8%. In the former case, we believe that these findings are cases where specifically the isotropic Gaussian assumption is violated, meaning the means of $D_c$ and $D_r$ are close together in comparison to the convex hull of these sets. We will add a discussion of these cases and additional visualizations in our appendix.

---

> > ### Comment · Reviewer_ADni · 2025-04-01
> >
> > After Rebuttal:
> > Thank you for the adding the analysis for Medical imaging. This new insight will help the medical imaging community to build more rigorous interpretability metrics. I strongly recommend this paper to be accepted at ICML. Also, i would request the authors to add the analysis in the appendix along with proper citations.

---

> > > ### Author Response · Authors · 2025-04-02
> > >
> > > We sincerely thank the reviewer for their constructive feedback and the strong recommendation. We will incorporate the analysis into our appendix with the proper citations and reference it in Section 4.1.
> > >
> > > We appreciate the reviewer's support.

---

### Official Review · Reviewer_CrYA · 2025-03-15

**Overall Recommendation:** 4

**Summary:**

The paper introduces a faster approximation to compute concept activation vectors. They do so through computing the mean vector for representations for the concept of interest and a random reference concept. The authors demonstrate that their approach is the same as the standard approach for computing concept activation vectors. However, through complexity analysis, the authors demonstrate that their algorithm runs faster. Experimentally, the authors demonstrate that their CAV method achieves the same or higher accuracy (compared to an SVM-based method), while reducing the computation time significantly.

## Update after rebuttal
My original score was a 4, and I am satisfied with their rebuttal; as such, I have decided to keep my score as a 4.

**Claims And Evidence:**

The authors make the following claims in their paper:
1. **Their fastCAV method is theoretically equivalent to SVM-based CAV under some assumptions** - This claim I believe is true. They show this by demonstrating that the expected decision boundary under a Gaussian setup exactly corresponds to the difference in means
2. **Their fastCAV method is significantly faster than the regular SVM-based CAV** - This claim is verified through their experiments, where they compare the computation time across different concepts in the Broaden dataset. Additionally, they verify such a claim theoretically through complexity analysis, where they demonstrate that their method (fastCAV) is faster by a factor of min(n,d) (compared to SVM-based approaches).
3. **Their fastCAV method maintains performance compared to regular SVM-based CAV** - They confirm this by comparing the accuracy of their predicted CAVs to separate the concepts in the random category from the concepts in the target category

**Essential References Not Discussed:**

I am not aware of any related work that is essential but not cited in the paper

**Experimental Designs Or Analyses:**

The experiments in Section 4 are reasonable; they compute the two most important quantities (accuracy and speed) and demonstrate that their method outperforms the baseline SVM-CAV for that. They also demonstrate that their fastCAV method can be applied to compute TCAV scores.

**Methods And Evaluation Criteria:**

The authors evaluate their fastCAV method through the Broaden dataset and ImageNet. The TCAV paper also uses the ImageNet dataset, and I believe that the particular dataset chosen is not of utmost importance (because CAV as a method can be used across datasets as long as concepts are labeled). Their methods are also reasonable for the task, as they demonstrate it's a simplified approximation to the overall CAV problem.

**Other Comments Or Suggestions:**

None

**Other Strengths And Weaknesses:**

The main strength of the paper is a new algorithm that is justified to quickly compute CAVs.

**Questions For Authors:**

1. What is the reason for considering pairwise similarity for robustness in Table 1

**Relation To Broader Scientific Literature:**

Their contributions help speed up a well known algorithm for computing CAVs, while maintaining accuracy.

**Theoretical Claims:**

The logic in Section 3.3 and Section 3.4 are reasonable, and I believe fairly intuitive.

---

> ### Author Rebuttal · Authors · 2025-03-31
>
> We appreciate the thoughtful feedback of the reviewer. We are glad that the logic in Sections 3.3 and 3.4 was judged as reasonable and intuitive. Similarly, we welcome the feedback regarding our experiments in Section 4. Particularly, the reviewer asked about our choice of pairwise cosine similarity, which we use to measure robustness. We would like to clarify the reasoning behind this decision and provide additional context.
>
> We consider pairwise similarity for robustness because it allows us to quantify the consistency and reliability of the concept activation vectors (CAVs) obtained from different methods. CAVs correspond to directions in a model’s activation space, which encode abstract concepts. Hence, for the decision boundary (Eq. (1)), we normalize the length of the CAV to unit length [Line 148, right-hand side]. Now, to compare the directions calculated with different methods for a given concept, we can directly compute the cosine of the angle, i.e., the cosine similarity, between the two vectors. Further, it is standard procedure to compute CAVs repeatedly against varying sampled random sets $D_r$ (Kim et al. 2018).
> In particular, pairwise similarity helps to evaluate robustness in two ways:
> 1. By calculating the pairwise similarity between FastCAV and SVM-based computation we can determine how closely aligned both methods are in identifying the direction corresponding to a concept.
> 2. By calculating the pairwise cosine similarity matrix for CAVs obtained from the same method while resampling $D_r$, we can assess the consistency of the concept directions across repetitions. Averaging over this similarity matrix provides a score that quantifies the robustness of the identified concept directions within a single method. A high average similarity score indicates that the method is producing consistent and reliable concept directions, which is essential for robustness.

---

> > ### Comment · Reviewer_CrYA · 2025-04-03
> >
> > Thank you for your response. I am happy with the paper, and will maintain my score

---

> > > ### Author Response · Authors · 2025-04-04
> > >
> > > We thank the reviewer and appreciate the positive assessment of our paper.

---

### Official Review · Reviewer_ULp4 · 2025-03-16

**Overall Recommendation:** 3

**Summary:**

The paper introduces fastcav, extending tcav to identify concept activation vectors by computing the mean of vectors and find the approximate direction of concept activation vectors to the concept space, which is assumed to be nearly orthogonal. The usage of mean reduces the dimensional complexity of using svm in tcav.

**Claims And Evidence:**

The major claim made in this paper is that fastcav is significantly faster than tcav, which is shown by the complexity analysis shown in section 3.4 and experiments in section 4.
Moreover, the claim that concept activation vectors computed through svm and fastcav are similar is also valided through comparing the cosine similarity of vector directions between svm and fastcav.

**Essential References Not Discussed:**

Not Discovered

**Experimental Designs Or Analyses:**

The experiments are mainly constructed to assess the accuracy and efficiency of identified cavs, which are sound, the usage of imagenet with multiple classes and multiple concept directions per neuron and acquisition of concepts from the widely tested broaden dataset makes the experiments valid and reliable.  A major drawback is that comparisions is only performed svm-cav and not with other concept based explanation approaches .

**Methods And Evaluation Criteria:**

The paper selects concepts from the broaden dataset and images from the imagenet dataset which are standard for the problem discussed. The proposed method signifantly reduces computational cost by eliminating need for svm training. The svm training is especially costly due to the presence of high dimensional intermediate feature representations.

**Other Comments Or Suggestions:**

Nil

**Other Strengths And Weaknesses:**

Nil

**Questions For Authors:**

How efficient and accurate fastcav is when compared to other concept explanation techniques?

**Relation To Broader Scientific Literature:**

The key contribution is the efficient computation of concept activation vectors, a popular concept based neural network explanation technique.

**Theoretical Claims:**

Not checked

---

> ### Author Rebuttal · Authors · 2025-03-31
>
> We thank the reviewer for their comments and thoughtful feedback. Specifically, we appreciate that the reviewer found our comparison between FastCAV and SVM-based computation valid and reliable. Concerning the lack of comparison to other CAV calculation methods, we note that we focused on SVMs following our theoretical analysis, which provides a connection to this existing approach. However, we agree with the reviewer that a comparison to other methods is beneficial to provide further evidence for our claims regarding the improved speed. Hence, we reran a smaller version of our first experiment (compare to our Table 1) and added the following approaches: logistic regression (Pfau et al. 2021, “Robust Semantic Interpretability: Revisiting Concept Activation Vectors”), sparsified logistic regression (Singla et al. 2021, “Using Causal Analysis for Conceptual Deep Learning Explanation”), Ridge classification as a faster alternative (Pedragosa et al. 2012, “Scikit-learn: Machine learning in python”), and LDA, which is closely related to our approach. Please excuse the simplified formatting due to markdown of the table presented here. We will ensure improved formatting in the updated paper.
>
> |Model|Method|Comp. Time [s]↓|Accuracy↑|Similarity↑|
> |---|---|---|---|---|
> |Inception-v3|FastCAV (Ours)|**0.013±0.020**|**0.950±0.059**|**0.826±0.011**|
> |Inception-v3|SVM|1.366±0.943|0.934±0.079|0.387±0.058|
> |Inception-v3|LDA|10.755±7.188|0.759±0.189|0.132±0.131|
> |Inception-v3|Logistic Regression|8.681±6.076|0.892±0.142|0.602±0.037|
> |Inception-v3|Sparse-Logistic Regression|8.648±5.960|0.895±0.141|0.603±0.036|
> |Inception-v3|Ridge|0.409±0.258|0.894±0.148|0.562±0.026|
> ||||||
> |ResNet50|FastCAV (Ours)|**0.016±0.019**|0.890±0.149|**0.791±0.030**|
> |ResNet50|SVM|2.908±2.440|0.871±0.147|0.398±0.062|
> |ResNet50|LDA|7.827±5.601|0.717±0.199|0.077±0.159|
> |ResNet50|Logistic Regression|8.375±5.509|0.921±0.115|0.652±0.042|
> |ResNet50|Sparse-Logistic Regression|8.474±5.687|0.924±0.113|0.654±0.040|
> |ResNet50|Ridge|0.314±0.225|**0.925±0.115**|0.602±0.026|
> ||||||
> |ConvNeXt-XXLarge|FastCAV (Ours)|**0.019±0.013**|0.923±0.081|**0.913±0.014**|
> |ConvNeXt-XXLarge|SVM|1.167±0.795|0.948±0.059|0.525±0.049|
> |ConvNeXt-XXLarge|LDA|16.842±11.911|0.895±0.149|0.521±0.105|
> |ConvNeXt-XXLarge|Logistic Regression|9.360±2.547|**0.962±0.051**|0.672±0.044|
> |ConvNeXt-XXLarge|Sparse-Logistic Regression|9.549±2.734|0.961±0.051|0.671±0.044|
> |ConvNeXt-XXLarge|Ridge|0.695±0.478|0.961±0.051|0.570±0.033|
> ||||||
> |RegNetY|FastCAV (Ours)|**0.039±0.017**|0.969±0.046|**0.825±0.009**|
> |RegNetY|SVM|5.207±1.054|**0.977±0.034**|0.128±0.066|
> |RegNetY|LDA|54.640±37.392|0.633±0.147|0.027±0.115|
> |RegNetY|Logistic Regression|11.529±4.791|0.964±0.052|0.697±0.063|
> |RegNetY|Sparse-Logistic Regression|11.747±4.917|0.963±0.053|0.698±0.062|
> |RegNetY|Ridge|2.184±1.300|**0.977±0.038**|0.675±0.020|
> ||||||
> |ViT-B/16|FastCAV (Ours)|**0.005±0.002**|0.827±0.129|**0.752±0.029**|
> |ViT-B/16|SVM|0.852±0.134|0.806±0.143|0.394±0.048|
> |ViT-B/16|LDA|3.421±0.443|**0.887±0.118**|0.556±0.121|
> |ViT-B/16|Logistic Regression|6.517±2.907|0.823±0.144|0.592±0.058|
> |ViT-B/16|Sparse-Logistic Regression|6.506±2.909|0.824±0.143|0.591±0.059|
> |ViT-B/16|Ridge|0.145±0.016|0.816±0.155|0.508±0.037|
> ||||||
> |ViT-H/14-CLIP|FastCAV (Ours)|**0.008±0.001**|0.875±0.153|**0.749±0.020**|
> |ViT-H/14-CLIP|SVM|0.456±0.078|0.872±0.151|0.342±0.048|
> |ViT-H/14-CLIP|LDA|8.867±1.037|0.823±0.174|0.551±0.149|
> |ViT-H/14-CLIP|Logistic Regression|7.091±2.658|**0.891±0.154**|0.621±0.025|
> |ViT-H/14-CLIP|Sparse-Logistic Regression|7.118±2.647|**0.891±0.154**|0.621±0.025|
> |ViT-H/14-CLIP|Ridge|0.288±0.013|0.889±0.158|0.595±0.023|
> ||||||
> |EVA-G/14|FastCAV (Ours)|**0.018±0.001**|0.890±0.147|**0.789±0.018**|
> |EVA-G/14|SVM|1.109±0.263|0.884±0.149|0.331±0.049|
> |EVA-G/14|LDA|19.135±2.200|0.809±0.211|0.418±0.070|
> |EVA-G/14|Logistic Regression|8.499±4.763|**0.903±0.148**|0.672±0.046|
> |EVA-G/14|Sparse-Logistic Regression|8.457±4.696|0.902±0.149|0.670±0.050|
> |EVA-G/14|Ridge|0.681±0.023|0.902±0.153|0.608±0.025|
> ||||||
> |EVA-02-L/14|FastCAV (Ours)|**0.024±0.001**|0.898±0.143|**0.814±0.020**|
> |EVA-02-L/14|SVM|1.524±0.594|0.897±0.155|0.291±0.054|
> |EVA-02-L/14|LDA|23.469±3.404|0.792±0.200|0.443±0.086|
> |EVA-02-L/14|Logistic Regression|7.659±4.693|0.909±0.148|0.685±0.052|
> |EVA-02-L/14|Sparse-Logistic Regression|7.693±4.731|0.909±0.147|0.688±0.050|
> |EVA-02-L/14|Ridge|0.923±0.019|**0.911±0.152**|0.606±0.023|
>
> Ridge classification offers a notable improvement in speed over SVM-based computation, our FastCAV still yields a substantial speed-up, while maintaining competitive performance. Further, we can see that vanilla LDA is not enough to achieve similar gains. Hence, we argue that the assumptions we make for FastCAV (compare Section 3.3) are necessary given practical considerations.

---

> > ### Comment · Reviewer_ULp4 · 2025-04-07
> >
> > Thank you for providing additional experimental comparisons with a broad set of CAV computation methods, including logistic regression, sparsified variants, Ridge classifiers, and LDA. I appreciate the effort taken to re-run experiments across multiple model architectures and the transparency in reporting both performance and computation time.
> >
> > The new results convincingly demonstrate that FastCAV provides a significant computational advantage while maintaining comparable or superior accuracy and concept similarity in most cases. The comparison with Ridge classification is particularly informative, as it offers a faster alternative to SVMs, yet FastCAV consistently outperforms even Ridge in terms of speed and often in similarity metrics as well. This reinforces your core claim about the practical utility of FastCAV in time-sensitive interpretability contexts. Overall, these additional results substantially improve the manuscript and address my initial concern regarding baseline comparisons.

---

> > > ### Author Response · Authors · 2025-04-07
> > >
> > > We thank the reviewer for their thoughtful and positive feedback.  Particularly, we are pleased that the additional experiments address the reviewer's concerns and significantly improve the manuscript. We will incorporate the results into our revision accordingly.

---

### Decision · Program_Chairs · 2025-05-01

**Decision:**

Accept (poster)

**Comment:**

There is no need to use SVMs when two groups of points are well-separated, as one can simply take the difference of the two barycenters as normal vector. Building on this idea, the paper proposes a re-implementation of the celebrated concept activation vectors, which runs orders of magnitude faster.

The reviewers are in general agreement that this is an interesting contribution to the field. I appreciated that the authors' answer contained many additional experiments which I would like to see in the revised version of the paper.